# Tumor site-directed A1R expression enhances CAR T cell function and improves efficacy against solid tumors

Kevin Sek [1,2,8] ✉, Amanda X. Y. Chen[1,2,8], Thomas Cole[1,2], Jesse D. Armitage[3,4], Junming Tong [1,2], Kah Min Yap [1,2], Isabelle Munoz [1,2], Phoebe A. Dunbar[1,2], Shiyi Wu[1,2], Marit J. van Elsas[1,2], Olivia Hidajat[1,2], Christina Scheffler [1,2], Lauren Giuffrida[1,2], Melissa A. Henderson[1,2], Deborah Meyran [1,2], Fernando Souza-Fonseca-Guimaraes [5], Dat Nguyen[1,2], Yu-Kuan Huang [1,2], Maria N. de Menezes [1,2], Emily B. Derrick [1,2], Cheok Weng Chan[1,2], Kirsten L. Todd [1,2], Jack D. Chan [1,2], Jasmine Li[1,2], Junyun Lai[1,2], Emma V. Petley[1,2], Sherly Mardiana[1,2], Anthony Bosco [6], Jason Waithman[3,4], Ian A. Parish [1,2], Christina Mølck[2], Gregory D. Stewart [7], Lev Kats [2], Imran G. House [1,2], Phillip K. Darcy [1,2,9] ✉ & Paul A. Beavis [1,2,9] ✉

The efficacy of Chimeric Antigen Receptor T cells against solid tumors is limited by immunosuppressive factors in the tumor microenvironment including adenosine, which suppresses Chimeric Antigen Receptor T cells through activation of the $A_{2A}$ receptor. To overcome this, Chimeric Antigen Receptor T cells are engineered to express $A_1$ receptor, a receptor that signals inversely to $A_{2A}$ receptor. Using murine and human Chimeric Antigen Receptor T cells, constitutive $A_1$ receptor overexpression significantly enhances Chimeric Antigen Receptor T cell effector function albeit at the expense of Chimeric Antigen Receptor T cell persistence. Through a CRISPR/Cas9 homology directed repair "knock-in" approach we demonstrate that Chimeric Antigen Receptor T cells engineered to express $A_1$ receptor in a tumor-localized manner, enhances anti-tumor therapeutic efficacy. This is dependent on the transcription factor *IRF8* and is transcriptionally unique when compared to $A_{2A}$ receptor deletion. This data provides a novel approach for enhancing Chimeric Antigen Receptor T cell efficacy in solid tumors and provides proof of principle for site-directed expression of factors that promote effector T cell differentiation.

Chimeric antigen receptor (CAR) T cell therapy involves genetic engineering of a patient's T cells to express a CAR that recognizes cell surface tumor antigens and drives potent T cell activation and subsequent killing of tumor cells. CAR T cells are an effective therapy for a range of hematological cancers achieving long-term anti-tumor durable responses[1,2]. However, in the context of solid tumors, which make up the majority of cancer cases, CAR T cell therapy remains inefficacious due to several additional challenges. These include but are not limited to T cell exhaustion and dysfunction, limited persistence and trafficking, and the immunosuppressive tumor microenvironment (TME)[3–5]. A major mediator of immunosuppression in the hypoxic TME is extracellular adenosine (eADO), a metabolite generated by the

ectoenzymes CD39, CD73 and CD38. eADO can bind to and signal through 4 subtypes of G-protein coupled adenosine receptors, the $A_1$ receptor ($A_1R$), $A_{2A}$ receptor ($A_{2A}R$), $A_{2B}R$ receptor ($A_{2B}R$) and $A_3$ receptor ($A_3R$), of which the $A_{2A}R$ is primarily expressed in T cells and responsible for suppression of CAR T cell function through activation of downstream adenylate cyclase and the localized accumulation of cyclic AMP (cAMP)[6]. Expression of $A_{2A}R$ is increased on activated T cells such as $T_H1$ and elevated on all subsets within the TME[7,8]. Critically, the $A_{2A}R$-cAMP axis is non-redundant, since the deletion of $A_{2A}R$ leads to abrogation of cAMP accumulation in T cells[6,9]. Seminal studies have demonstrated that targeting the $A_{2A}R$ axis can potently enhance T cell function and anti-tumor responses, where $A_{2A}R$ pharmacological blockade could augment the anti-tumor activity of immune checkpoint blockade through enhanced CD8$^+$ T cell responses[10,11]. These encouraging results have led to the evaluation of inhibitors targeting the $A_{2A}R$ pathway, upstream CD73, CD39 or CD38 receptors or the hypoxia-HIF1$\alpha$ axis in preclinical and clinical studies to treat a range of cancers[12–18]. Targeting the adenosine pathway has also been shown to enhance the efficacy of adoptively transferred T cells recognizing tumor cells through either their TCR or CAR[9,19–22].

Recently, we demonstrated that CRISPR/Cas9 deletion of $A_{2A}R$ in CAR T cells abrogated cAMP accumulation and potently enhanced function[9]. The effect observed was superior compared to pharmacological blockade or knockdown shRNA approaches, thus highlighting the feasibility and advantages of gene-editing approaches to targeting adenosine immunosuppression[9]. Given that suppression of T cell function by G$\alpha$s coupled $A_{2A}R$ was mediated by cAMP accumulation, we therefore sought to target the $A_{2A}R$-cAMP axis through the expression of the alternative signaling G$\alpha$i coupled adenosine receptors, $A_1R$ or $A_3R$ that have previously been described to suppress cAMP production in other cell types[17]. $A_1R$ is not naturally expressed on T cells and whilst $A_3R$ is expressed at only low levels[23,24] the use of $A_3R$ agonists has been shown to suppress cAMP in resting T cells and enhance T cell anti-tumor functionality[25,26]. We hypothesized that $A_1R$ or $A_3R$ expression in CAR T cells could not only prevent increased cAMP due to $A_{2A}R$ signaling in the eADO rich TME, but potentially lead to inverse signaling and therefore enhance CAR T cell function against cancers with immunosuppressive TME[27–30].

In this work, we show that expression of $A_1R$ but not $A_3R$ suppresses cAMP production in CAR T cells and enhances both murine and human CAR T cell function by promoting their production of IFN$\gamma$ and TNF, cytokines that have been demonstrated to be critical for anti-tumor efficacy of CAR T cells in solid tumors[21,31,32]. However, constitutive $A_1R$ expression leads to a loss of the 'stem-like' memory population (CD45RO$^-$CD45RA$^+$CD62L$^+$CD27$^+$) and reduces the persistence of CAR T cells in vivo limiting overall therapeutic efficacy, which is consistent with previous studies that have shown that less differentiated CAR T cells have greater self-renewal capacity and ability to mediate anti-tumor control[2,33–36]. Therefore, we developed an approach to restrict $A_1R$ expression to the tumor site to preserve persistence and drive increased anti-tumor function through enhanced effector differentiation in response to the CAR antigen.

CRISPR/Cas9 mediated homology directed repair (HDR) was used to knock-in $A_1R$ under the control of the NR4A2 promoter, which we have previously shown to be highly effective in restricting transgene expression to the tumor site[37]. This resulted in enhanced anti-tumor efficacy and function relative to control CAR T cells and was associated with superior persistence compared to constitutive overexpression of $A_1R$ in CAR T cells. The enhanced effector function of $A_1R$ expression was transcriptionally associated with profound epigenetic and transcriptional changes targeting distinct pathways relative to $A_{2A}R$ deletion in CAR T cells. Unsupervised network analysis led to the identification of an $A_1R$ dependent gene module, within which *IRF8* was identified to be a key transcription factor. Indeed, CRISPR/Cas9 mediated deletion of *IRF8* reversed the phenotype elicited by $A_1R$

expressing CAR T cells, indicating a downstream role for *IRF8* in $A_1R$ signaling and a previously unappreciated role for *IRF8* in promoting effector CAR T cell differentiation. Taken together these results suggest that tumor-localized expression of $A_1R$ leads to enhanced CAR T cell efficacy and holds potential for improving the treatment of solid tumors, particularly those with adenosine-rich immunosuppressive microenvironments. Furthermore, this work is proof of principle that tumor-directed expression of factors that promote T cell differentiation may be leveraged to enhance CAR T cell function.

## Results

### $A_1R$ expression enhances anti-tumor functionality and effector differentiation of mouse and human CAR T cells

Murine CAR T cells targeting human HER2 were generated using retroviral transduction as per our previous studies and engineered to express either $A_1R$ or $A_3R$ linked to a truncated NGFR selection marker[21,38–40] (Fig. 1A). Transduction efficiencies were similar between groups and ranged between 40–60%, which could be enriched to >90% using magnetic bead isolation (Supplementary Fig. 1A–B). Overexpression of $A_1R$ or $A_3R$ was confirmed by qRT-PCR, whilst $A_{2A}R$ gene expression was unchanged following $A_1R$ or $A_3R$ overexpression (Fig. 1B and Supplementary Fig. 1C). $A_1R$ and $A_3R$ mRNA were unaltered by CAR stimulation ($\alpha$TAG), confirming that an overexpression strategy could effectively modulate the expression levels of these receptors (Supplementary Fig. 1D). Given that canonical $A_1R$ and $A_3R$ signaling results in decreased cAMP levels, functional expression of $A_1R$ or $A_3R$ was assessed via the measurement of cAMP production following stimulation with eADO or the pan-adenosine receptor agonist, NECA. As expected, activation of $A_{2A}R$ with increasing concentrations of the pan adenosine receptor agonist NECA (Supplementary Fig. 1E) or eADO (Fig. 1C) resulted in the accumulation of cAMP. This effect was completely abrogated in CAR T cells generated from $A_{2A}R^{-/-}$ mice, confirming the dominance of the $A_{2A}R$ in this effect. $A_1R$ expression led to a significantly increased EC$_{50}$ for NECA and eADO to generate cAMP by ~4–8 fold, supporting our hypothesis that $A_1R$ expression can inhibit $A_{2A}R$ mediated cAMP production (Supplementary Fig. 1F). However, no change in cAMP concentrations were observed following $A_3R$ expression in CAR T cells, possibly related to the lower affinity of adenosine/NECA for the $A_3R$ relative to the $A_1R$ or $A_{2A}R$[41]. To assess the impact of $A_1R$ or $A_3R$ expression on anti-tumor function, $A_1R$ or $A_3R$ CAR T cells were cocultured with E0771-Her2 or 24JK-Her2 tumor cells. Strikingly, $A_1R$ but not $A_3R$ CAR T cells produced significantly higher levels of both IFN$\gamma$ and TNF when cocultured with either tumor cell line (Fig. 1D). Whilst the use of the $A_1R$ antagonist DPCPX confirmed that these effects were $A_1R$ mediated (Supplementary Fig. 1G), neither $A_1R$ nor $A_3R$ CAR T cells were protected from $A_{2A}R$ mediated suppression, verified through $A_{2A}R$ blockade with SCH58261 antagonist (Supplementary Fig. 1H), which is likely explained by the dominant effect of the $A_{2A}R$ on cAMP levels (Fig. 1C). $A_1R$ overexpression did not significantly enhance cytotoxic function as determined by overnight $^{51}$Cr release killing assays (Supplementary Fig. 1I) indicating that the dominant phenotype was enhanced cytokine production. Having established that $A_1R$ expression enhances the function of murine anti-Her2 CAR T cells, we proceeded to investigate this in the human context by expressing $A_1R$ in human CAR T cells targeting the Lewis Y antigen, which are the subject of ongoing phase I clinical trials (NCT03851146)[42]. CAR T cells were transduced with $A_1R$ linked to the NGFR marker gene and sorted for CAR and transgene expression (Supplementary Figs. 2A–B). Due to the lack of reliable flow cytometry antibodies for the hA$_1$R, a short Myc-tag sequence was engineered onto the extracellular region of the wild-type hA$_1$R to confirm protein expression on the cell surface (Supplementary Fig. 2C, and Supplementary Table 1). Co-expression of Flag-tag CAR and Myc-tag hA$_1$R was determined by flow cytometry, confirming successful cell surface expression of $A_1R$ on anti-Lewis Y CAR T cells (Supplementary Figs. 2C–D). Consistent with observations in the

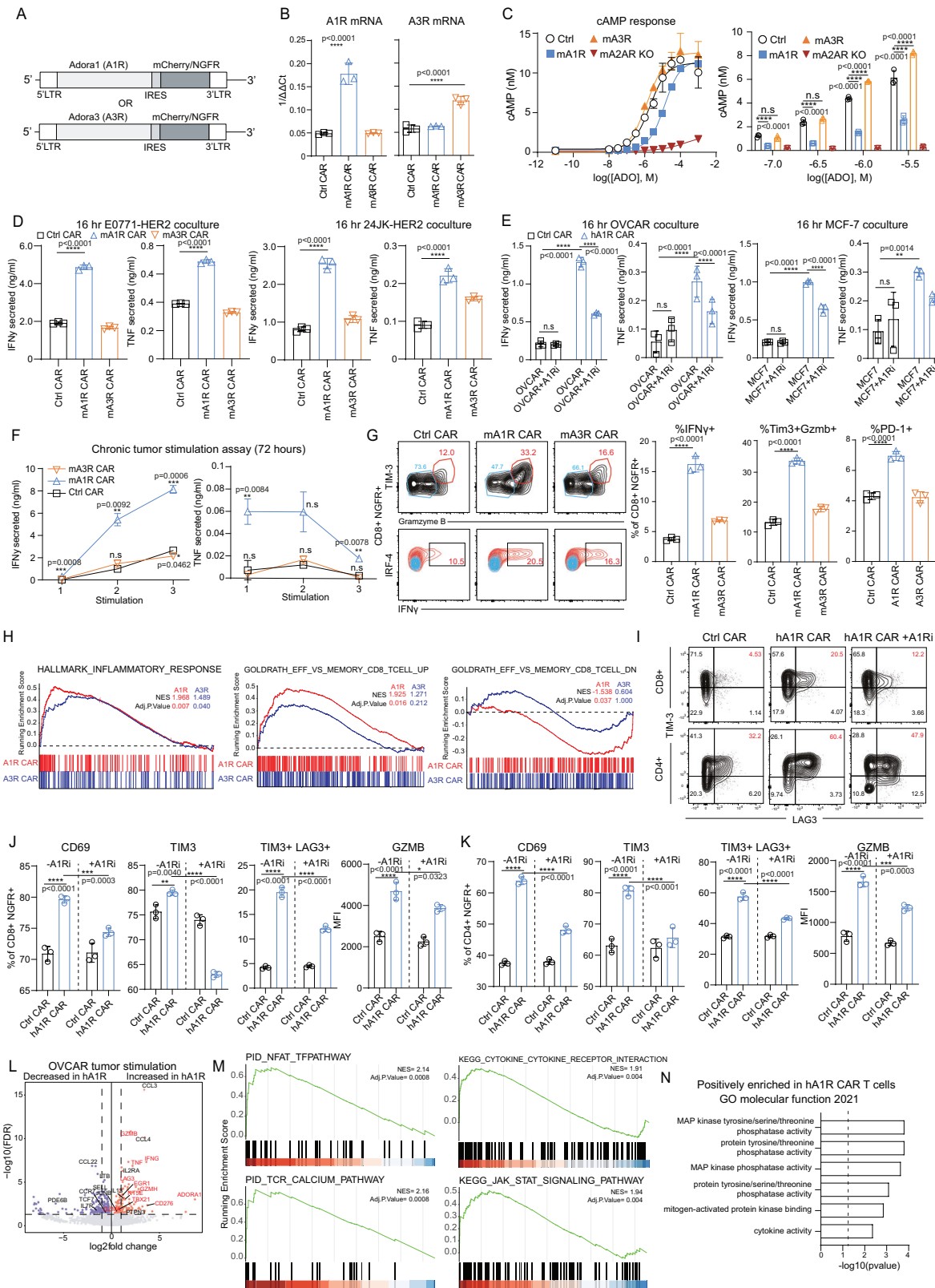

murine system, overexpression of hA$_1$R on human anti-Lewis Y CAR T cells enhanced the production of IFNγ and TNF by CAR T cells upon coculture with either OVCAR-3 or MCF7 (both Lewis Y$^+$) tumor cells (Fig. 1E). This effect was partially reversible following the addition of the A$_1$R antagonist DPCPX (OVCAR-3 + A$_1$Ri) to tumor cell cocultures, confirming that direct A$_1$R signaling was required for this phenotype (Fig. 1E).

CAR T cells gradually lose their capacity to secrete cytokines and kill tumor cells with serial antigen stimulation due to exhaustion or dysfunction[31,43,44]. We next investigated the impact of A$_1$R signaling on the effector functionality and differentiation of murine CAR T cells in serial antigen cocultures with tumor cells where murine anti-Her2 CAR T cells were serially challenged with tumor cells with multiple rounds of stimulation over 72 h. To investigate the impact of A$_1$R signaling on

**Fig. 1 | A$_1$R expression enhances anti-tumor functionality and effector differentiation of mouse and human CAR T cells. A** Schematic of mouse or human adenosine receptors, A$_1$R and A$_3$R, tagged with mCherry/NGFR expression markers. **B** qRT-PCR cycle threshold normalized against Rpl32, $n = 3$ independent experiments. **C** cAMP response to ADO of mouse CAR T cells primed with FSK (1 μM), $n = 2$ independent experiments. **D** Cytokine production in mouse anti-HER2 CAR T cells after coculture with E0771-HER2 or 24JK-HER2 tumor cell lines for 16 h. **E** Cytokine production by human anti-Lewis Y CAR T cells after coculture with OVCAR-3 or MCF-7 tumor cell lines for 16 h in the presence or absence of an A$_1$R antagonist (DPCPX, 100 nM). **F** Cytokine production by mouse anti-HER2 CAR T cells after serial coculture with E0771-HER2 tumors every 24 h. **G** FACS plots and quantification of CAR T cells after 72-hour serial coculture. **H** GSEA plot of pathways from the mSigDB database based on RNA-seq of tumor stimulated A$_1$R or A$_3$R CAR T cells versus Control. **I–K** Human anti-Lewis Y A$_1$R CAR T cells stimulated with OVCAR-3 tumor cells for 16 h in the presence or absence of an A$_1$R inhibitor (100 nM), DPCPX. **I** FACS plots, **J–K** marker expression in CD8$^+$/CD4$^+$ CAR T cells. **L** Differentially expressed genes of OVCAR-3 stimulated human anti-Lewis Y CAR T cells. **M** GSEA of PID and KEGG datasets of stimulated hA$_1$R CAR T cells versus control. **N** EnrichR plots of differentially expressed genes in both mouse and human A$_1$R CAR T cells versus control for positively enriched genesets. $P$ values were determined using the Fisher's exact test/hypergeometric test based on the enrichR package. ****$p < 0.0001$, ***$p < 0.001$, **$p < 0.01$, *$p < 0.05$, mean ± SD (**B–E, G, J, K**). One-way ANOVA or (**F**) two-way ANOVA. **D, F, G, H** All mouse data is representative of at least 3 independent experiments, with the bulk RNA-seq experiment performed with 2 technical replicates. **E, I–L** All human data is representative of at least 3 independent experiments. Human bulk RNA-seq was performed with 3 technical replicates. **H, M** GSEA $P$-value estimation is based on an adaptive multi-level split Monte-Carlo scheme based off the fgsea package.

effector differentiation, CAR T cells were generated using IL-7 and IL-15 to maintain CAR T cells in a less differentiated phenotype and mimic cytokine preconditioning standardly given to CAR T cells prior to in vivo transfer[45]. Following coculture with E0771-Her2 tumor cells, A$_1$R CAR T cells exhibited significantly increased production of IFNγ and TNF (Fig. 1F) and increased expression of effector related genes PD-1 (*Pdcd1*), TIM-3 (*Havcr2*), *Gzmb* and *Irf4* compared to control and A$_3$R CAR T cells (Fig. 1G). Consistent with cytokine secretion assays, A$_1$R CAR T cells expressed higher levels of IFNγ as detected by intracellular staining (Fig. 1G).

To provide further mechanistic insight into the mechanism by which A$_1$R expression enhanced effector function of CAR T cells RNA-sequencing was performed after serial antigen stimulation. Notably A$_1$R cells exhibited a significant enrichment for the Hallmarks inflammatory response pathway (NES 1.968) and effector versus memory differentiation in CD8 T cells (GSE1000002; NES 1.92) indicating that A$_1$R CAR T cells were more effector-like[46] (Fig. 1H). We also observed significant upregulation of genes encoding components of the NFAT/AP-1 signaling pathway heterodimers *Junb* and *Fos*, as well as NFAT associated transcription factors *Egr1* and *Egr2*, and *Tox* in both unstimulated and tumor stimulated A$_1$R CAR T cells[47–49] (Supplementary Fig. 2E). In human anti-Lewis Y CAR T cells cocultured with OVCAR-3 tumors, we similarly observed increased activation in A$_1$R CAR T cells as demonstrated by an increased proportion of TIM-3, LAG-3, CD69 and Granzyme B positive cells (Fig. 1I-K). These phenotypes were partially reversed with short-term A$_1$R inhibition, indicating that both acute and serial A$_1$R stimulation contributed to this phenotype (Fig. 1I–K). To further interrogate this, bulk RNA-seq analysis was performed on human anti-Lewis Y CAR T cells cocultured with OVCAR-3 tumors. Analyses of differentially expressed genes (DEGs) between stimulated hA$_1$R CAR T cells and controls showed increased expression of the effector molecules *IFNG*, *TNF*, granzymes and checkpoint receptors *LAG3*, *NT5E* (CD73) and *CD276* (B7-H3). Similar to observations in mouse A$_1$R CAR T cells, *JUNB* and transcription factors TBET *(TBX21)* and *EGR1* were also upregulated in human A$_1$R CAR T cells (Fig. 1L). Notably, hA$_1$R expression led to a significant enrichment for genes associated with cytokine, JAK-STAT, NFAT and calcium signaling (Fig. 1M). Comparing significant DEGs between mouse and human A$_1$R CAR T cells stimulated by tumor cells showed a high level of overlap cross-species, with A$_1$R significantly reducing expression of memory markers *S1PR1* or *TCF7*, while increasing expression of dual specificity phosphatases (*DUSP2/5*), *EGR1*, granzymes and cytokines (Supplementary Fig. 2F–G). We performed enrichR gene signature enrichment analyses (GSEA) and protein-protein association analyses (STRING) on significant DEGs that overlap between mouse and human A$_1$R CAR T cells (Supplementary Figs. 2F–G). Notably, A$_1$R expression led to a significant enrichment for genes associated with MAPK signaling, which is consistent with the canonical role of A$_1$R in inducing MAPK pathway activation in other cell types (Fig. 1N)[50].

## Constitutive A$_1$R expression drives exhaustion of CAR T cells and reduction in T-stem-like memory populations

We next assessed the anti-tumor efficacy of mouse A$_1$R and A$_3$R CAR T cells in vivo by adoptive transfer to treat E0771-Her2 tumor-bearing mice utilizing a syngeneic Her2-Tg model previously characterized by our lab[21,38,40]. Whilst anti-Her2 CAR T cells were effective in significantly reducing tumor growth, there was no enhanced efficacy observed following A$_1$R expression (Fig. 2A). Analysis of spleens, tumors and draining lymph nodes (dLN) at day 7 post therapy showed that although tumor infiltrating A$_1$R CAR T cells exhibited a more effector phenotype as demonstrated by increased expression of granzyme B ($p = 0.06$), Tim-3 and PD-1 (Supplementary Fig. 3A), CAR T cell numbers were significantly reduced relative to control CAR T cells within the spleen and tumor (Fig. 2B). Previous studies by both our group and others have shown that a less differentiated T stem-cell memory (T$_{SCM}$) phenotype in CAR T cells confers greater long-term engraftment post adoptive transfer, and notably this population was reduced with constitutive A$_1$R expression (CD62L$^+$CD44$^{dim}$) (Supplementary Fig. 3B)[45,51–54].

Similarly, A$_1$R overexpression in human anti-Lewis Y CAR T cells also led to a significant reduction in the T$_{SCM}$ subsets in both CD4$^+$ and CD8$^+$ CAR T cells, defined by CD45RA$^+$CD45RO$^-$CD62L$^+$CD27$^+$ cells as previously described[53,54] (Figs. 2C, and Supplementary Fig. 3C–D). Furthermore, there was a reduction in CD62L$^+$CD69$^-$ and concomitant increase in CD62L$^-$CD69$^+$ fractions in both CD4$^+$ and CD8$^+$ A$_1$R CAR T cells (Supplementary Fig. 3E). These results suggested that constitutive A$_1$R expression leads to increased effector T cell differentiation. Consistent with this, we observed that A$_1$R expression significantly enhanced the expression of CD69 and LAG3 on CAR T cells even in the absence of antigen stimulation (Fig. 2D, and Supplementary Figs. 2D, and 3F). Critically, the loss of T$_{SCM}$ subsets, and effector T cell differentiation phenotypes (increased CD69$^+$CD62L$^-$ or LAG3$^+$CD69$^+$ fractions) could be reversed with the A$_1$R antagonist DPCPX (Fig. 2D, Supplementary Figs. 3D–F). This data suggests that constitutive A$_1$R signaling drives T cell activation and consistent with this hypothesis, CD69 was also increased in mouse CD8$^+$ A$_1$R CAR T cells prior to antigen stimulation (Supplementary Fig. 3G). To further interrogate the impact of A$_1$R expression on T cell differentiation in the absence of antigen we analyzed the transcriptome of control and A$_1$R CAR T cells in this context. A$_1$R CAR T cells exhibited significantly increased expression of genes involved in the NFAT/AP-1 pathway (*JUNB*), NFAT associated transcription factors (*TOX*, *TOX2*, *EGR1*) and checkpoint receptors (*LAG3*, *CD276*), indicating that constitutive A$_1$R expression led to profound changes in CAR T cell biology even in the absence of CAR stimulation (Fig. 2E). GSEA analyses identified a positive enrichment for signatures associated with T cell receptor and cytokine signaling as well as a positive and negative enrichment for a T cell exhaustion and memory signature respectively[55] (Fig. 2F). To identify the A$_1$R transcriptional signature agnostic of CAR stimulation or T cell activation, we

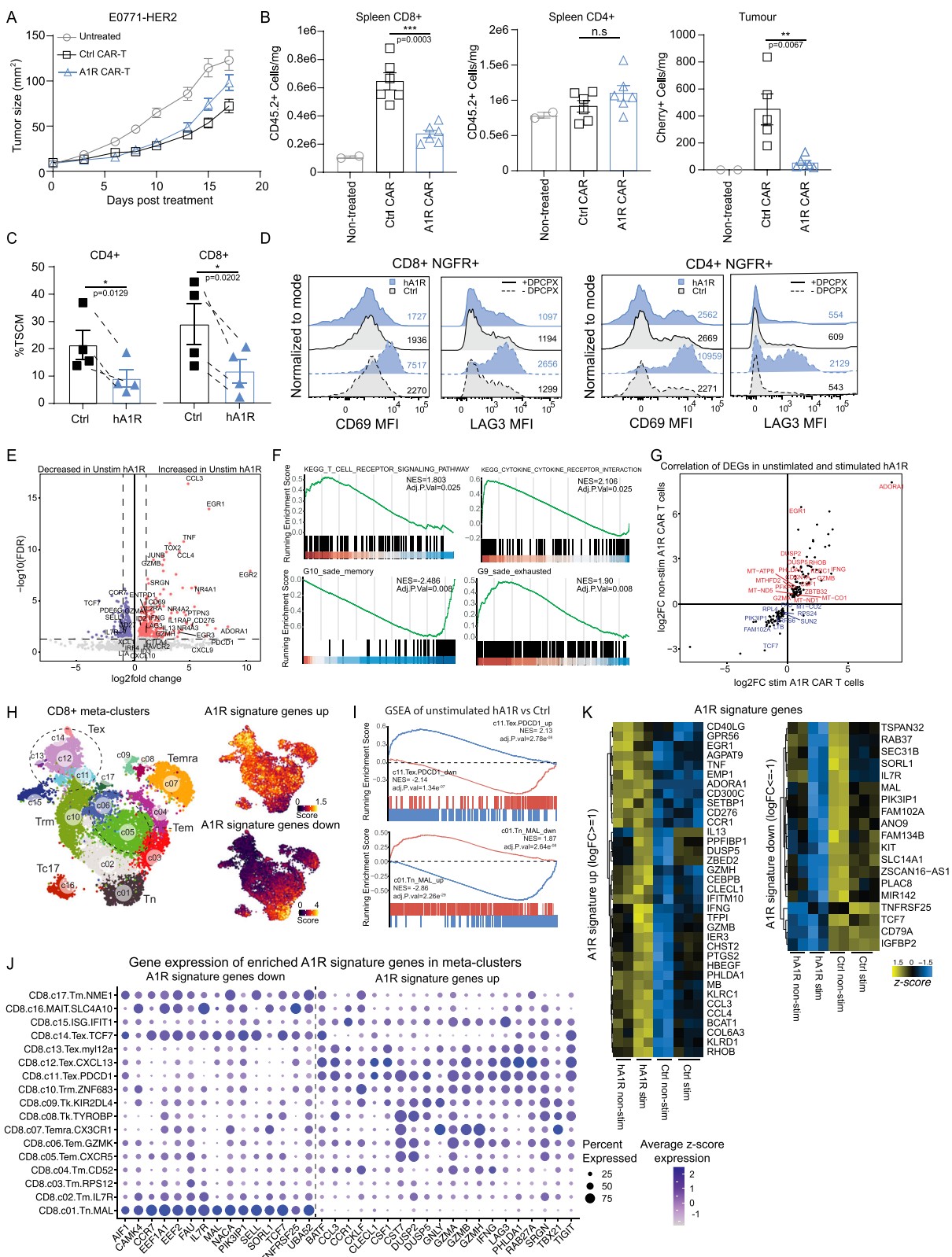

correlated genes differentially modulated in both stimulated and unstimulated $A_1R$ CAR T cells versus control CAR T cells (Fig. 2G). Comparing the $A_1R$ gene signature to meta-clusters across 21 cancer types from a pan-cancer atlas, showed a strong positive enrichment for exhaustion associated meta-clusters (c11) and negative enrichment for naïve (c01) meta-clusters (Figs. 2H, I, and Supplementary Table 2)[56]. Interestingly, these Tex meta-clusters consist of IFNγ and

granzyme producing T cells, with high levels of genes from the $A_1R$ signature (Fig. 2J, and Supplementary Fig. 4A). Conversely, genes negative regulated by $A_1R$ were expressed in the Tn naïve clusters (Fig. 2J, and Supplementary Fig. 4B). A heatmap of the genes in the $A_1R$ signature showed that many of these genes associated with effector function and memory were modulated prior to CAR stimulation (Fig. 2K).

**Fig. 2 | Constitutive A$_1$R expression drives terminal differentiation of CAR T cells.** C57BL6-HER2 transgenic mice bearing orthotopic E0771-HER2 tumors were pre-treated with 4 Gy of total body irradiation prior to adoptive transfer of 2 doses of 10e$^6$ anti-HER2 CAR T cells. 5 doses of 50,000 IU of IL-2 were injected i.p every 24 h. **A** Tumor growth curve, representative of 2 independent experiments with $n$ = 4-6 mice per group. **B** CAR T cell counts in spleen and tumors from n = 5-6 mice except for non-treated group ($n$ = 2 mice). Data represented as the mean ± SEM (**A**, **B**). **C** Quantification of %T$_{SCM}$ (CD45RO$^-$CD45RA$^+$CD27$^+$CD62L$^+$) in CD4$^+$ and CD8$^+$ NGFR$^+$ human anti-Lewis Y CAR T cells shown as pooled data of means ± SEM from $n$ = 4 independent healthy donors. **D** CD69 and LAG3 expression in human CAR T cells cultured with or without A$_1$R antagonist DPCPX (1 μM) in CD8$^+$/CD4$^+$ CAR T cells prior to tumor stimulation, representative of $n$ = 4 donors. **E** Volcano plot of DEGs for unstimulated CAR T cells. **F** GSEA of DEGs enriched in A$_1$R CAR T cells using the KEGG and Sade-Feldman single-cell RNA-seq datasets (GSE120575).

**G** Correlation of significant DEGs between tumor stimulated and unstimulated hA1R vs Ctrl CAR T cells. **H** CD8$^+$ T cell meta-clusters from integrated atlas of single cell RNA-seq experiments across 21 tumor types and cells from 316 donors[56]. Derived A$_1$R signature genes up or down-regulated are overlayed onto meta-clusters, **I** GSEA of up or down-regulated genes from each meta-cluster for unstimulated hA$_1$R CAR T cells vs control. **J** Quantification of gene expression in each meta-cluster GSEA for select core-enriched genes from the A$_1$R signature as defined by ref. 56. **K** Heatmap of significant DEGs from the A$_1$R signature upregulated or downregulated with a logFC≥1 or logFC≤1 respectively between tumor stimulated and unstimulated hA$_1$R vs Ctrl CAR T cells. ****$p$ < 0.0001, ***$p$ < 0.001, **$p$ < 0.01, *$p$ < 0.05. **C** paired two-sided t-test (**B**) One-way ANOVA or (**A**) two-way ANOVA. Human bulk RNA-seq was performed on a single donor with 3 technical replicates. **F**, **I** GSEA *P*-value estimation is based on an adaptive multi-level split Monte-Carlo scheme based off the fgsea package.

These analyses supported the notion that A$_1$R expression could accelerate T cell differentiation, but this was not coupled to antigen-mediated activation and so led to premature differentiation of CAR T cells and a consequent failure to persist. Therefore, to negate the tonic signaling effects of A$_1$R driving effector differentiation prior to adoptive transfer, we pre-conditioned A$_1$R CAR T cells with the DPCPX inhibitor prior to adoptive transfer to OVCAR-3 tumor bearing mice. However, we found no enhanced therapeutic benefit compared to control CAR T cells (Supplementary Fig. 4C), and this was likely due to reactivation of A$_1$R signaling after adoptive transfer resulting in reduced persistence of CD8$^+$ A$_1$R CAR T cells in vivo (Supplementary Fig. 4D).

**CAR T cells engineered to express A$_1$R under the control of the NR4A2 promoter exhibit enhanced CAR T cell function without loss of T-stem-like memory subsets**

While long-term DPCPX pre-conditioning A$_1$R CAR T cells with DPCPX did not fully restore in vivo persistence, we did observe that it led to even greater production of IFNγ and TNF by A$_1$R CAR T cells versus no pre-conditioning (Fig. 3A). This highlighted the potential for controlled A$_1$R expression to drive greater T cell activation and differentiation in an acute setting, and so we reasoned that inducible A$_1$R expression may overcome limitations with persistence and loss of the T$_{SCM}$ subset (Supplementary Fig. 5A). To this end, we utilized a CRISPR knock-in approach pioneered by our laboratory[37] to drive inducible expression of A$_1$R upon CAR antigen stimulation at the tumor site. This approach leverages the regulatory mechanisms of endogenous genes that are exclusively expressed by CAR T cells within the tumor site. Specifically, expression of transgenes via the endogenous NR4A2 promoter leads to stringent tumor-localized expression (Fig. 3B). NR4A2 is upregulated by T cells upon activation and therefore enables A$_1$R expression to be coupled to antigen stimulation. Furthermore, it is important to note that this approach simultaneously inserts the A$_1$R gene while knocking out NR4A2 (Supplementary Fig. 5A). To highlight the potential for this approach to couple transgene expression to CAR activation, we first cocultured anti-Lewis Y CAR T cells engineered to express NGFR under the control of the NR4A2 promoter with Lewis Y$^+$ tumor cells. In line with our previous work, NGFR expression was only observed following CAR activation and chronic stimulation (Supplementary Figs. 5B, C) and assessment of A$_1$R mRNA in NR4A2/A$_1$R engineered CAR T cells confirmed that A$_1$R expression was only observed post CAR activation (Fig. 3C). To confirm that antigen-mediated control of transgene expression was maintained in vivo, we adoptively transferred NR4A2/ NGFR human CAR T cells into OVCAR-3 tumor bearing mice and observed tumor-site specific expression of NGFR (Supplementary Figs. 5D, E).

Having demonstrated that the CRISPR knock-in approach could successfully link A$_1$R expression to CAR T cell activation in vitro, we tested our hypothesis that NR4A2/A$_1$R engineered CAR T cells would exhibit enhanced effector functions without loss of "stem-like" memory populations associated with constitutive A$_1$R expression.

Indeed, NR4A2/A$_1$R engineered CAR T cells exhibited significantly increased expression of IFNγ, TNF and IL-2 secretion, primarily in CD8$^+$ CAR T cells, following tumor cell coculture (Fig. 3D, E, and Supplementary Fig 6A). Importantly, unlike constitutively expressing A$_1$R CAR T cells, NR4A2/A$_1$R CAR T cells were phenotypically indistinguishable from control CAR T cells prior to activation with no significant changes to the proportion of T$_{SCM}$ memory T cells or expression of LAG3 and CD69 (Fig. 3F, G). Inducible A$_1$R expression drove the upregulation of LAG-3, TIM-3 and PD-1, and differentiation into an effector state with reduced CD62L expression, which could be reversed by A$_1$R blockade in CD8$^+$ CAR T cells (Fig. 3H, and Supplementary Fig. 6B). To evaluate the broad applicability of NR4A2/ hA$_1$R to enhance T cell adoptive therapy, we also applied this engineering approach to CARs targeting the HER2 and ROR1 antigens and in the context of CARs with either a CD28 or 4-1BB signaling domain. Consistent with our results obtained with anti-Lewis Y CAR T cells, NR4A2/A$_1$R engineered anti-HER2 CAR T cells exhibited significantly enhanced cytokine production upon coculture with HER2$^+$ MCF-7 breast tumors (Fig. 3I, and Supplementary Fig. 6C). Regardless of costimulatory domains expressed, HER2-CARs with CD28 or 41BB domains secreted enhanced cytokine production against MDA-MB231 breast tumors (Supplementary Figs. 6D, E). Similarly, NR4A2/hA$_1$R engineering enhanced the cytokine production of anti-ROR1 CAR T cells when cocultured with MDA-MB231 tumors in vitro (Supplementary Fig. 6F). Taken together, we propose that A$_1$R expression enhances CAR T cell cytokine production irrespective of the type of CAR expressed or antigen target/ level of expression.

To further interrogate the underlying biology of NR4A2/hA$_1$R engineered CAR T cells we performed bulk ATAC-seq of NR4A2/hA$_1$R CAR T cells after OVCAR-3 tumor stimulation which demonstrated clear epigenetic changes in NR4A2/A$_1$R engineered CAR T cells that were only apparent following activation (Fig. 3J, K). Motif enrichment analyses of differentially accessible peaks showed an enrichment for transcription factors associated with T cell activation and effector differentiation, consistent with our hypothesis that A$_1$R expression led to more robust T cell differentiation[57,58] (Fig. 3L). Next, bulk RNA-sequencing of NR4A2/hA$_1$R CAR T cells from a serial stimulation assay was performed, which revealed that NR4A2/hA$_1$R engineered CAR T cells became progressively more distinct and enriched for the A$_1$R gene signature compared to control or NR4A2KO CAR T cells following each round of stimulation (Figs. 3M−O, and Supplementary Fig. 6G). With each round of stimulation and induction of the A$_1$R signature, we observed an enrichment for gene signatures involved with T cell cytokine signaling and inflammatory response pathways and enrichment for cytotoxic and exhaustion signatures versus either control (Fig. 3P) or NR4A2 KO CAR T cells (Fig. 3Q, and Supplementary Fig. 6H). Critically, the downregulation of memory signatures was not significant prior to stimulation or with initial activation, and was only significantly down regulated with repeat stimulation, highlighting the potential of this approach for maintaining memory while driving

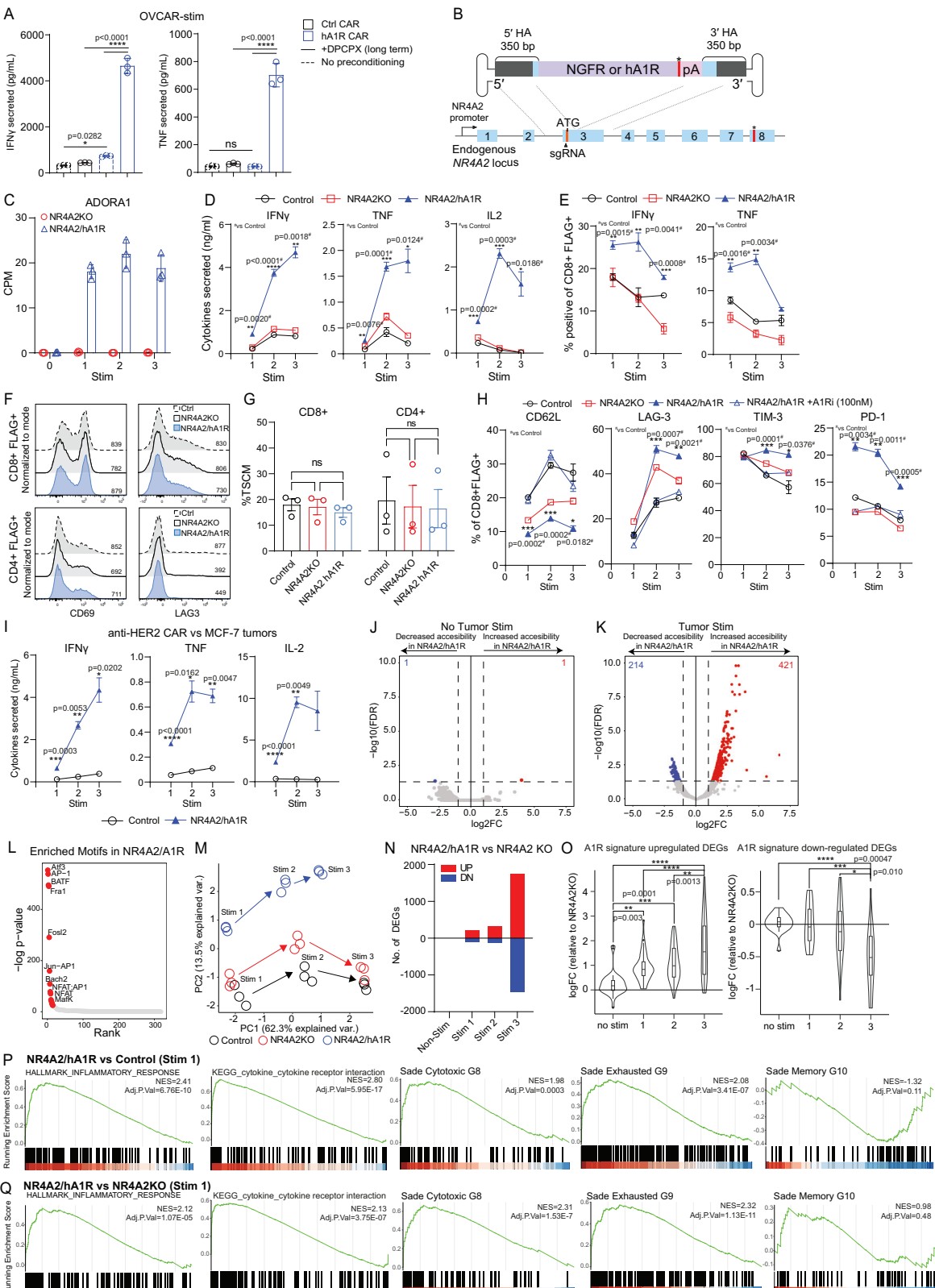

effector differentiation or function in vivo (Fig. 3P, Q, andSupplementary Fig. 6H).

## NR4A2/hA₁R engineered CAR T cells elicit enhanced therapeutic efficacy against solid tumors

To determine the in vivo efficacy of NR4A2/hA₁R engineered CAR T cells, 5x10⁶ anti-HER2 CAR T cells or 15×10⁶ anti-Lewis Y CAR T cells were adoptively transferred to treat MDA-MB231 breast (Fig. 4A) or OVCAR-3 ovarian (Fig. 4B) tumor bearing mice respectively. Strikingly in each instance, NR4A2/A₁R engineered CAR T cells exhibited significantly enhanced therapeutic efficacy and led to tumor eradication in 50% of OVCAR-3 tumor bearing mice (Fig. 4B–D), which was not associated with any signs of toxicity or weight loss (Supplementary Fig 7A-B). Critically, in contrast to our previous data with constitutively

**Fig. 3 | CAR T cells engineered to express A$_1$R under the control of the NR4A2 promoter exhibit enhanced CAR T cell function without loss of memory precursors. A** Cytokine secretion of anti-Lewis Y CAR T cells that were pre-conditioned in culture with DPCPX (100 nM) prior to co-culture with OVCAR-3 tumors, representative with mean ± SD of triplicate cultures. **B** CRISPR/HDR template with homology arms for the NR4A2 gene, to knock-in payload transgene under the control of the NR4A2 promoter. **C** RNA-seq counts per million (CPM) of ADORA1 gene after serial coculture with OVCAR-3 tumors, represented as the mean ± SD (**D**) Cytokine secretion and **E** intracellular cytokine staining in CAR T cells after serial coculture with OVCAR-3 tumors, represented as the mean ± SD of triplicate cultures. **F** CD69 and LAG3 expression in CD8$^+$/CD4$^+$ CAR T cells without tumor stimulation. **G** Quantification of %T$_{SCM}$ (CD45RO$^-$CD45RA$^+$CD27$^+$CD62L$^+$) in CD4$^+$/ CD8$^+$ human anti-Lewis Y CAR T cells shown as pooled data of means ± SEM from 3 donors. **H** Expression of markers following coculture with OVCAR-3 tumor cells, represented as the mean ± SD of triplicate cultures. **I** Cytokine secretion by anti-

HER2 CAR T cells against MCF-7 breast cancer after serial coculture, represented as the mean ± SD of triplicate cultures. **J–K** Differential accessibility peaks from ATAC-sequencing in **J** unstimulated or (**K**) OVCAR-3 tumor stimulated human CAR T cells. **L** HOMER motif analyses of enriched motifs in NR4A2/hA$_1$R CAR T cells. **M** PCA plot for each round of coculture. **N** Quantificaiton of DEGs with each round of stimulation. **O** Magnitude of up or downregulated DEGs from the previously identified A$_1$R signature in NR4A2/hA$_1$R CAR T cells. Box-plot defined by box (1$^{st}$ and 3$^{rd}$ interquartile range), median-line, whiskers extending to points within 1.5x interquartile range. **P–Q** GSEA of gene-sets from mSigDB Hallmarks, KEGG and Sadefeldman gene signatures (GSE120575) for **P** hA$_1$R vs Control or (**Q**) hA$_1$R vs NR4A2KO CAR T cell groups after first stimulation with tumor cells. ****$p < 0.0001$, ***$p < 0.001$, **$p < 0.01$, *$p < 0.05$. **A, F, G, K, O** One-way ANOVA or (**D, E, H, I, L, M, N**) two-way ANOVA. **C, J–Q** Human bulk ATAC-seq and RNA-seq was performed in independent experiments, 3 technical replicates each.

expressing A$_1$R CAR T cells (Fig. 2B), the numbers of NR4A2/hA$_1$R CAR T cells were equivalent to controls in both the spleen and the tumor at up to day 28 post therapy (Fig. 4E–G). Furthermore, NR4A2/hA$_1$R engineered cells exhibited a similar frequency of T$_{SCM}$ CAR T cells in the spleen (Fig. 4H), indicating that linking A$_1$R expression to CAR T cell activation at the tumor site overcame the deleterious effects on persistence. Unsupervised ex vivo analysis of tumor-infiltrating CAR T cells by flow cytometry led to the identification of 7 clusters of CD8$^+$FLAG$^+$ CAR T cells (Fig. 4I–K). NR4A2/hA$_1$R engineered CAR T cells exhibited an increased emergence of a population of cells with higher expression of CD69, PD-1 and reduced expression of TCF-1, CD45RA and CD27 (Fig. 4J, K; **cluster 3**). NR4A2/A$_1$R CAR T cells also demonstrated a concomitant reduction of a cluster of cells that expressed TCF1 and CD62L (Fig. 4J, K; **cluster 6**). We hypothesized that this could represent the loss of a memory like population associated with enhanced differentiation following A$_1$R expression. Indeed, further analysis of tumor infiltrating T exhausted (Tex; PD-1$^{Hi}$TCF-1$^-$), T progenitor exhausted CD62L$^{+/-}$(Tpex; PD-1$^{INT/LOW}$TCF-1 + ) and T effector (Teff; PD-1$^{INT/LOW}$TCF-1$^-$) subsets (Supplementary Fig. 7C) revealed a significant increase in Teff subsets and a corresponding decrease in both CD62L$^+$ and CD62L$^-$ Tpex subsets (Figs. 4L, and Supplementary Fig. 7D) in NR4A2/hA$_1$R engineered CAR T cells, while no significant change in memory subsets were observed in the spleen[34]. Altogether, these results indicated that tumor-localized expression of A$_1$R led to significantly enhanced therapeutic effects, which was associated with enhanced effector cell differentiation of CAR T cells specifically at the tumor site.

## A$_1$R expression and A$_{2A}$R deletion target distinct transcriptional pathways to enhance CAR-T cell efficacy

While we initially sought to target the A$_{2A}$R-cAMP axis through the expression of the alternative signaling Gαi coupled A$_1$R, our results suggested that A$_1$R activation and A$_{2A}$R blockade may be acting independently. Notably A$_1$R expression led to enhanced cytokine production in the absence of exogenous adenosine whereas our previous work suggested that A$_{2A}$R knockout CAR T cells elicited enhanced cytokine production only in the presence of adenosine-mediated suppression[9]. To evaluate this further a direct comparison of CRISPR A$_{2A}$R knockout and NR4A2/A$_1$R knock-in was performed using anti-Lewis Y CAR T cells. These head-to-head experiments clearly demonstrated that A$_1$R expression drove enhanced production of IFNγ and TNF upon activation with OVCAR-3 tumors, which was not observed with A$_{2A}$R deletion alone (Supplementary Fig. 8A). The adenosine analogue, NECA suppressed IFNγ and TNF in control CAR T cells, but that this effect was lost in A$_{2A}$R KO CAR T cells (Supplementary Fig. 8A). Interestingly, while NECA suppressed IFNγ in NR4A2/A$_1$R CAR T cells, overall cytokine produced remained higher for all three cytokines tested (Supplementary Fig. 8A). Next, we demonstrated that increased cytokine production could be reversed with A$_1$R antagonist (DPCPX),

which showed no such effects on control or A$_{2A}$R KO CAR T cells (Supplementary Fig. 8B).

To further interrogate potential differences, tumor stimulated NR4A2/A$_1$R and A$_{2A}$RKO CAR T cells treated with or without NECA were analyzed by RNA-seq to determine the transcriptional impacts of A$_1$R and A$_{2A}$R deletion. Principal component analyses of sequenced samples identified a clear separation mainly across Principal Component 1 (PC1, 64.1% variance explained) between NR4A2/hA$_1$R CAR-T cells compared to A$_{2A}$RKO or NR4A2KO controls (Fig. 5A). NECA treatment led to separation mainly across Principal Component 2 (PC2, 12.8% variance explained) in both NR4A2/hA$_1$R and NR4A2KO CAR-T cells, which was not observed in A$_{2A}$RKO CAR T cells, highlighting that NECA was mainly acting on the A$_{2A}$R in both NR4A2/hA$_1$R and NR4A2KO controls (Fig. 5A). Furthermore, few genes were differentially expressed between A$_{2A}$RKO and NR4A2KO control cells in the absence of NECA (18 DEGs up/down), whereas A$_1$R knock in led to major transcriptional changes (1683 DEGs up/ 1809 DEGs down) versus NR4A2KO controls (Fig. 5B). In the presence of NECA during tumor stimulation, there was very little overlap between transcriptional changes across NR4A2/hA$_1$R, A$_{2A}$RKO and NR4A2KO CAR-T cells (Fig. 5B), highlighting the distinct transcriptional pathways associated with A$_1$R signaling. Finally, we demonstrated that the A$_1$R gene signature identified previously was only significantly enriched in NR4A2/hA$_1$R but not A$_{2A}$RKO CAR-T cells, suggesting that A$_1$R expression and A$_{2A}$R deletion target distinct transcriptional pathways to enhance CAR-T cell efficacy (Fig. 5C).

## Enhanced effector function of A$_1$R expressing CAR T cells is dependent on *IRF8*

To identify key transcriptional programs and molecular drivers involved in downstream A$_1$R signaling that underpin the enhanced CAR-T cell phenotype, unsupervised Weighted Gene Correlation Network analysis (WGCNA) was performed on the serial stimulation bulk RNA-seq dataset (Fig. 3). WGCNA identified 9 co-expression networks or modules based on sets of genes which were significantly correlated across all groups (Figs. 5D, and Supplementary Fig. 9A). Of these, one module (denoted Red) was of particular interest given that it was significantly increased in NR4A2/A$_1$R engineered CAR T cells at all time points (Fig. 5E). The Red module comprised of 221 genes and was strongly correlated with the A$_1$R signature and T cell activation and cytokine signatures (Fig. 5F, G). The majority of the Red module genes were associated with T cell effector function and differentiation, such as effector molecules (LTA, GZMB), type 2 interferon cytokines (IFNG) and receptors (IFNGR1), chemokines (CXCL9, CCL3, CCL4) and transcription factors (TBX21, EGR1, EGR3, IRF8) (Fig. 5G). GSEA of the Red module showed enrichment for gene ontology (GO) signatures linked to positive regulation of cytokine, chemokine production and signaling (Fig. 5H). To identify transcription factors that potentially act as molecular drivers of the Red module, we ranked them based on

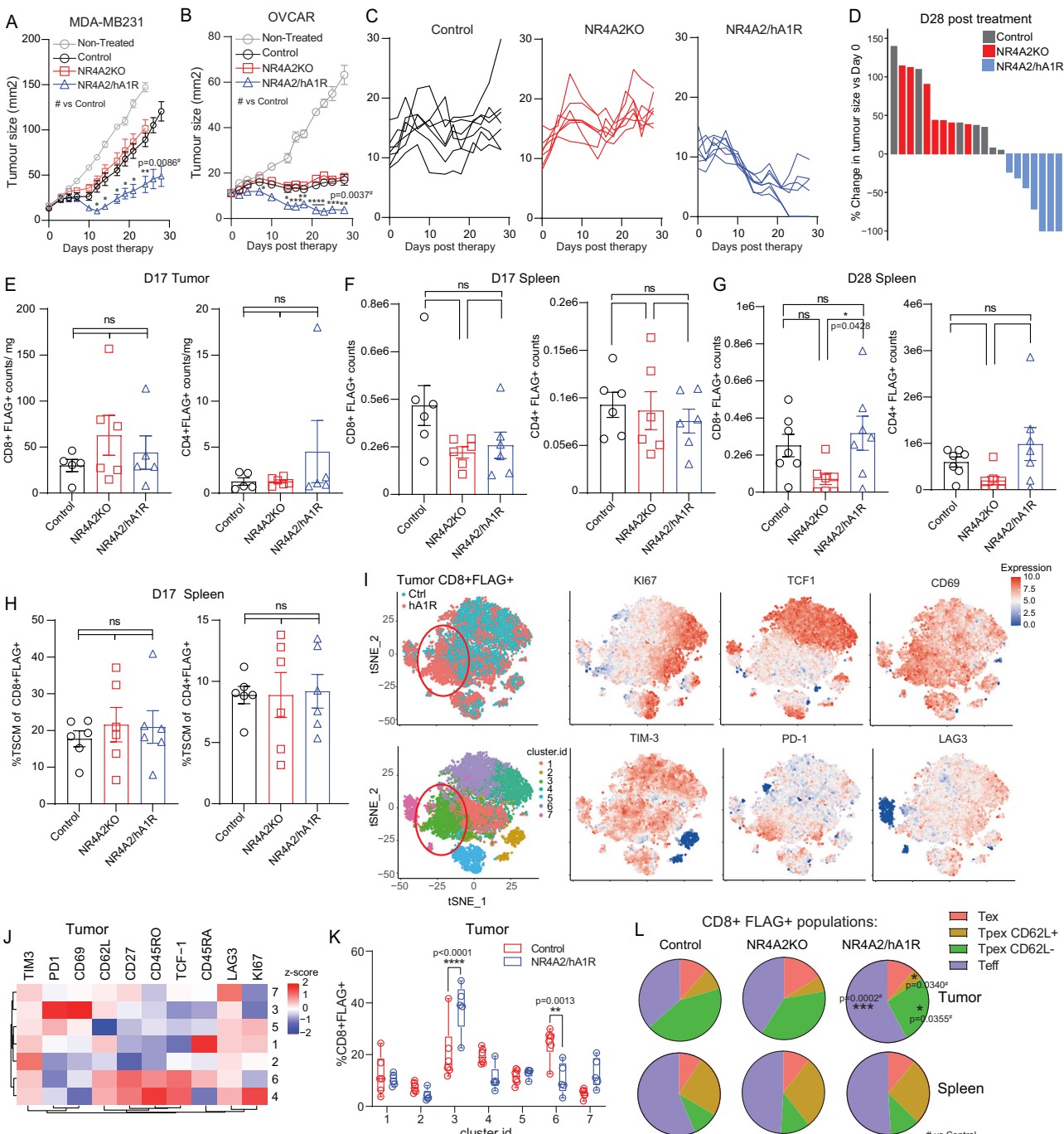

**Fig. 4 | Tumor site specific expression of hA₁R enhances CAR T cell therapeutic efficacy against solid tumors in vivo.** NSG mice were injected in the fourth mammary fat pad or sub-cutaneously with $1.25 \times 10^6$ HER-2$^{Low}$ MDA-MB231 breast or $5 \times 10^6$ LewisY$^+$ OVCAR-3 ovarian tumor cells respectively. Once tumors were established (15-20 mm²), mice were then irradiated (1 Gy) and treated with one dose of $5 \times 10^6$ anti-HER2 or $15 \times 10^6$ anti-Lewis Y CAR T cells, respectively. Mice were supplemented with 50,000 IU of IL-2 on days 0-4 post treatment. **A** Tumor growth, data shown as means ± SEM of $n = 5–6$ mice per group (**B**) Tumor growth, data shown as means ± SEM of $n = 7$ mice per group and **C** individual tumor growth curves. **D** Waterfall plot of % change in tumor size on Day 28 post therapy. **E–G** Counts of CD8$^+$ and CD4$^+$ FLAG$^+$ CAR T cells in **E** tumor and spleen at (**F**) D17 post therapy and **G** D28 post therapy. **H** Quantification of %T$_{SCM}$ CAR T cells in spleen D17 post therapy. **B–H** Data represented as the mean ± SEM of $n = 6$ mice per group. **I** TSNE plots and unbiased clustering of tumor infiltrating CD8$^+$FLAG$^+$ CAR T cells analyzed by flow cytometry at day 17 post therapy. **J** Heatmap and **K** quantification of marker expression sorted by unbiased clustering in tumor at D17 post therapy. Box-plot defined by box (interquartile range, 1$^{st}$ and 3$^{rd}$ quartiles), median-line, whiskers extending to SD. **L** Quantification of memory subsets of CD8$^+$ FLAG$^+$ T$_{PEX}$ (PD-1$^{INT}$TCF-1$^+$CD62L$^{+/-}$), T$_{EFF}$ (PD-1$^{INT}$TCF-1$^-$) and T$_{EX}$ (PD-1$^{HI}$TCF-1$^-$) in tumor and spleen at D17 post therapy. ****$p < 0.0001$, ***$p < 0.001$, **$p < 0.01$, *$p < 0.05$. **E–H, L** one-way anova or (**A, B, K**) two-way ANOVA. **A, B** Statistics shown are NR4A2/hA₁R versus NR4A2KO groups. Data representative of 2 (a, $n = 5–6$ mice per group) or 3 independent experiments (**B**, $n = 7$ mice per group).

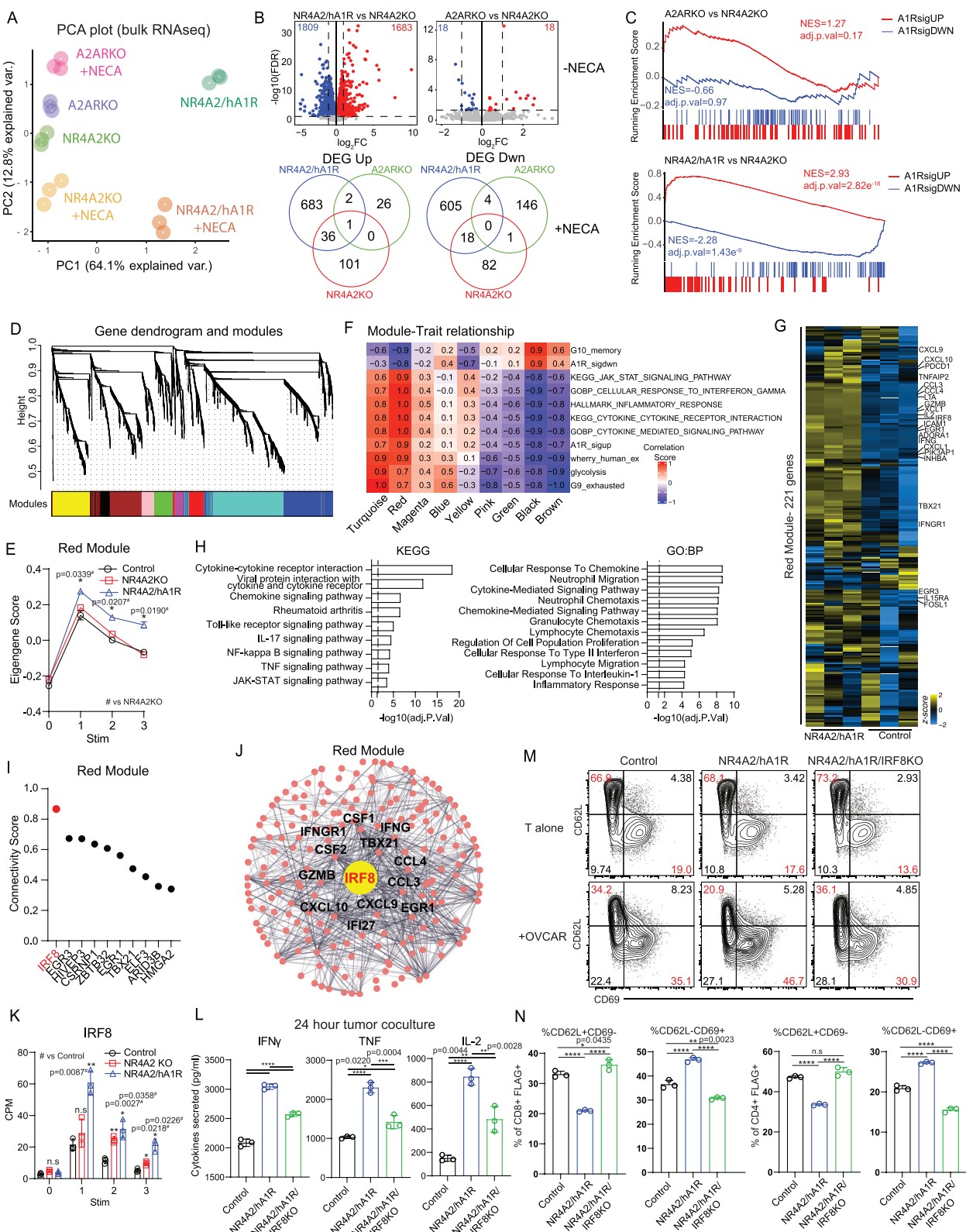

intramodular connectivity, and we found that *IRF8* was the highest ranked transcription factor (Fig. 5I, J) and was also significantly increased in NR4A2/A₁R engineered CAR T cells at each timepoint (Fig. 5K). While multiple studies have examined *IRF8* to have important roles in the context of myeloid cells and its tumor suppressor function through downstream STAT1 signaling, there has been limited studies of the role of *IRF8* in cytotoxic T cells[59,60]. Miyagawa et al. and others

have linked *IRF8* expression with T cell receptor signaling, CD8+ T cell effector differentiation or exhaustion[44,61], while a more recent study identified *IRF8* expression as a biomarker predicting greater CD8+ T cell activation, infiltration and response to monoclonal antibody therapy in ER-negative breast cancer patients[62]. Therefore, we decided to investigate further the role of *IRF8* in enhancing T cell function downstream of A₁R signaling. CRISPR-mediated deletion of *IRF8*

**Fig. 5 | Enhanced effector function of A₁R expressing CAR T cells drives *IRF8* dependent distinct transcriptional pathways compared to A₂ₐR deletion.** Bulk RNA-seq was performed on anti-Lewis Y CAR T cells stimulated for 72 h with OVCAR-3 tumors in the presence or absence of pan-adenosine receptor agonist, NECA (10 μM). **A** PCA plot, (**B**) DEGs for of CAR T cells stimulated with tumors in the absence of NECA (top). Venn Diagram showing overlap of up or down-regulated DEGs between NECA vs no NECA treated CAR T cells (bottom). **C** GSEA of A₁R gene signature for A₂ₐRKO and NR4A2/hA₁R CAR T cells versus NR4A2KO controls. **D** Network dendrogram and gene-coexpression modules identified using WGCNA of human CAR T cells after serial coculture with OVCAR-3 tumors over 72 h. **E** Eigengene score of individual groups at each round of stimulation for the red module, two-way ANOVA, n = 3 technical replicates per group, data represented as the mean ± SD. **F** Module-trait relationship of key gene signatures against modules eigengene scores. **G** Heatmap of 221 genes in the red module after 1 round of

coculture. **H** GSEA of the genes identified from the red module using EnrichR. *P* values were determined using the Fisher's exact test/ hypergeometric test based off the enrichR package. **I** Ranked connectivity scores for transcription factors within the red module (**J**) Network plot of *IRF8* and 1st neighbors in the red module. **K** CPM counts of *IRF8* at each stimulation timepoint, represented by mean ± SD of triplicates. **L–N** *IRF8* knockout NR4A2/hA₁R CAR T cells were generated using CRISPR/Cas9 editing and stimulated with OVCAR-3 tumor cells overnight. **L** Cytokine secretion, **M, N** activation (CD69⁺CD62L⁻) and memory (CD62L⁺CD69⁻) phenotype. **L–N** Data shown as means ± SD of triplicate cultures and representative of 2 independent experiments with 2 healthy donors. ****p < 0.0001, ***p < 0.001, **p < 0.01, *p < 0.05. **G, K** two-way ANOVA or (**L, N**) one-way ANOVA. Human bulk RNA-seq was performed on a single donor with 3 technical replicates. **C** GSEA *P*-value estimation is based on an adaptive multi-level split Monte-Carlo scheme based off the fgsea package.

(Supplementary Fig. 9B) in control human anti-Lewis Y CAR T cells did not significantly modulate cytokine production or expression of memory and exhaustion markers TCF-1, TIM-3 or PD-1 in unstimulated or tumor stimulated CAR T cells (Supplementary Figs. 9C–E). However, deletion of *IRF8* partially reversed the enhanced cytokine production mediated by NR4A2/A₁R CAR T cells and reversed the emergence of CD62L⁻CD69⁺ NR4A2/hA₁R CAR T cells relative to controls when cocultured with OVCAR-3 or MCF-7 tumors (Fig. 5L–N). Given that the deletion of *IRF8* had no impact on cytokine production in control CAR T cells (Supplementary Fig. 9C), and the significantly upregulation in NR4A2/hA₁R CAR T cells relative to control cells (Fig. 5K), it is likely that *IRF8* has a unique and central role in promoting cytokine production downstream of A₁R signaling.

## Discussion

To date there are over 190 clinical trials of CAR T cell therapies currently active or completed, with over 600 trials currently being recruited. Whilst the use of anti-CD19 CAR T cells to treat refractory CD19⁺ B-ALL has been incredibly effective, with up to 90% of pediatric patients achieving complete responses, CAR T cells have thus far been less efficacious in the context of solid tumors with a complete response rate of only 4.1% in a meta-analysis of solid tumor trials[63,64]. This has been attributed to a myriad of barriers to overcome in solid tumors which include limited persistence exhaustion/dysfunction of CAR T cells, tumor heterogeneity and the immunosuppressive TME[3–5]. Adenosine-mediated immunosuppression is a major immune checkpoint for tumor-infiltrating CAR T cells, with the ectoenzymes CD73 and CD39, responsible for eADO production found to be over-expressed in multiple cancers and prognostic of poor patient outcomes[27,29,65–67], and limiting both conventional chemo and immunotherapy treatments[6,17]. Therefore, there is great interest in developing drugs targeting the eADO-axis, with small molecular antagonists and antibodies against CD73, CD38 and CD39 under clinical development[12–18,68,69]. Our previous studies demonstrated that while A₂ₐR blockade with small molecular antagonist could enhance the in vivo efficacy of CAR T cells, gene-targeting approaches utilizing CRISPR-Cas9 to delete the A₂ₐR led to significantly superior efficacy and selectivity relative to pharmacological blockade[9]. To date, there are limited studies looking to use gene-engineering approaches to target adenosine signaling in CAR T cells and given the success in 'armoring' CAR T cells against the TME through the gene-deletion of suppressive A₂ₐR, NR4A family members, or overexpression of effector factors such c-Jun or BATF, we hypothesized that overexpression of the alternative signaling A₁R or A₃R would make a good candidate for protecting CAR T cells against eADO immunosuppression[9,44,57]. Unlike the A₂ₐR, A₁R and A₃R are Gαi/o coupled and inhibit downstream adenylate cyclase and cAMP accumulation, which we hypothesized could be leveraged to reverse the immunosuppressive effect of adenosine on T cells mediated by A₂ₐR signaling. Whilst A₁R is not expressed on lymphocytes physiologically, previous studies have

suggested a role for A₃R in promoting T cell effector functionality. For example, Morello and Montinaro et al. described anti-tumor effects from A₃R agonist drugs, which were attributed to enhanced cytokine production by T cells in both adoptive transfer and single agent treatment in preclinical tumor models[25,26].

In CAR T cells, our data demonstrates that overexpression of A₁R but not A₃R could suppress cAMP accumulation and enhance cytokine and granzyme production in response to tumor antigen. Activation and exhaustion markers PD-1, TIM-3, LAG-3 and CD69 were upregulated in both mouse and human A₁R CAR T cells. This was associated with transcriptional programs associated with T cell effector differentiation and interestingly an increase in the expression of genes related to MAPK phosphatase pathways and calcium signaling. This alludes to T cell activation via phospholipase C (PLC-β/γ) hydrolysis of membrane-bound PIP₂ to generate inositol triphosphate (IP3) and diacylglycerol (DAG). Both secondary messengers can then trigger calcium (Ca²⁺) dependent calcineurin and NFAT pathways and drive other signaling pathways including protein kinase C (PKC)[70].

While A₁R dramatically enhances CAR T cell function in vitro, we found that constitutive A₁R expression led to a loss of persistence once these cells were transferred in vivo. Indeed, A₁R expression led to a reduction in the T stem cell memory compartment (T_SCM) prior to stimulation during the manufacturing phase and this was due to antigen independent A₁R tonic signaling driving activation and effector differentiation which could be reversed by A₁R pharmacological blockade. The loss of this T_SCM compartment was likely responsible for the failure of constitutive A₁R expression to enhance therapeutic activity given that the frequency of T_SCM cells in the CAR T cell product has been demonstrated to correlate with improved therapeutic efficacies due to increased long-term persistence[2,53,71,72]. Whilst the effects of A₁R expression were unexpected prior to our study, this effect is consistent with a link between the adenosine pathway that has been observed by both our group and others[73]. We previously showed that A₂ₐR knockdown using shRNA similarly depleted the T_SCM population of the CAR T cell product[9] whilst conversely it was recently shown that ADA overexpression could enhance the proportion of T_SCM through the conversion of adenosine to inosine[74].

Nevertheless, our results indicated that A₁R signaling likely needed to be transient to enhance CAR T cell efficacy. Indeed, we found that pre-conditioning constitutive A₁R CAR T cells with the A₁R antagonist, DPCPX, could preserve the T_SCM subset and further enhance T cell activation and cytokine production upon antigen stimulation. To restrict A₁R expression to the tumor site we therefore utilized a CRISPR knock-in approach to drive A₁R expression under the control of the endogenous NR4A2 promoter, which we have previously demonstrated to be highly tumor-restricted in its transcriptional activity. We hypothesized that this approach would enable A₁R expression to be restricted to the tumor site, effectively driving effector differentiation and enhanced anti-tumor function by CAR T cells without any impact on persistence and memory.

This engineering strategy successfully enabled CAR T cell activation specific expression of $A_1R$ upon tumor antigen-encounter. Transcriptional and epigenetic analyses confirmed that NR4A2/$A_1R$ CAR T cells were almost identical to control CAR T prior to stimulation and then rapidly became more activated after CAR stimulation. Importantly, this approach restored $T_{SCM}$ subsets relative to controls, and enhanced cytokine production and effector differentiation in in-vitro coculture assays. NR4A2/$A_1R$ engineered CAR T cells elicited significantly enhanced CAR T cell efficacy in vivo and led to enhanced CAR T cell differentiation at the tumor site without any impact on persistence in the spleen or tumor. Together, the data supports a model whereby $A_1R$ site-specific expression drives T cell activation and differentiation into a highly cytotoxic T effector subset within the tumor but not the spleen, which in turn augments CAR T cell anti-tumor efficacy without any impact on persistence and expansion in vivo. Critically, this approach targets different pathways compared to $A_{2A}R$ deletion, given our observations of unique transcriptional pathways targeted by $A_1R$ expression. Critically, $A_{2A}R$ deletion does not affect persistence as opposed to $A_1R$ signaling, indicating that both pathways could potentially be targeted simultaneously to enhance CAR-T cell efficacy.

Finally, to identify downstream mechanisms driving the $A_1R$ response, unbiased co-expression network analysis was performed on bulk RNA-sequencing data from NR4A2/h$A_1R$ CAR T cells. Network connectivity analyses implicated the *IRF8* transcription factor in playing an important role downstream of $A_1R$ signaling. Subsequent deletion of *IRF8* in NR4A2/h$A_1R$ engineered CAR T cells abrogated the enhanced cytokine production in both serial and long-term stimulation assays and suppressed $A_1R$ driven enhanced tumor killing during the second round of long-term restimulation. This was associated with reduced CAR T cell activation and increased expression of memory markers. *IRF8* has been shown previously to be upregulated by IFNα, IL-12 and STAT-4 in NK cells playing a critical role in driving proliferation and viral control but thus far has not been implicated downstream of adenosine receptor signaling in T cells[75,76]. Taken together, this demonstrates a previously unrecognized and important role of *IRF8* downstream of $A_1R$ signaling in mediating enhanced CAR T cell cytokine production and effector differentiation. Additionally, IRF factors have been implicated in metabolic processes as stress sensors, like other factors such as Foxo1, which could be manipulated to further improve CAR-T cell persistence[77,78]. In further studies it will be interesting to delineate the mechanism by which *IRF8* enhances the effector function of CAR T cells in this context given there are a limited number of studies that have evaluated the role of *IRF8* in T cells.

In summary, we have demonstrated that overexpression of $A_1R$ in CAR T cells leads to enhanced cytokine production and effector differentiation at the expense of loss of long-lived memory subsets. To overcome this, we utilized a CRISPR/Cas9 knock-in strategy to drive $A_1R$ expression specifically at the tumor site, leading to enhanced therapeutic efficacy and comparable engraftment, persistence, and expansion. Several studies have demonstrated the feasibility of favorably controlling CAR T cell differentiation through controlled programming to promote anti-tumor function, most of which have focused on promoting stemness features through inosine[74] or cytokine preconditioning or overexpression of memory associated transcription factors such as FOXO1[78,79]. Others have targeted factors that drive or prevent exhaustion such as c-Jun[57], c-Myb[80] or BATF[44]. This highlights the importance of preserving memory but enhancing effector differentiation, and is to our knowledge, the first demonstration of an approach to engineer CAR T cell differentiation states specifically at the tumor site. Importantly this approach can be applied to any receptor or transcription factor that promotes short-term CAR T cell functionality at the expense of long-term persistence. Moreover, our work reveals a novel approach and mechanism in which to armor CAR T cells against adenosine mediated suppression in immunosuppressive solid tumors, which holds significant translational potential given that these CRISPR/Cas9 editing approaches are consistent with those currently used clinically for CAR T cells[81,82].

## Methods

### Animal models and Cell lines
The C57BL/6 mouse 24JK (Patrick Hwu, NIH, Bethesda, Maryland, USA)[83] and the breast carcinoma cell line E0771 (Robin Anderson, Peter MacCallum Cancer Centre)[84] were engineered to express truncated human HER2 as previously described[85]. Human OVCAR-3 ovarian cancer and MCF7 and MDA-MB231 breast cancer cell lines were obtained from the American Type Culture Collection. Short tandem repeat (STR) analyses were used to confirm the identity of the cell lines, which were used within 10 passages of a master stock to ensure their accuracy. Cells were cultured in Roswell Park Memorial Institute (RPMI) media (Gibco Life Technologies) supplemented with 10% heat-inactivated fetal bovine serum (FBS), 1 mM sodium pyruvate, 2 mM glutamine, 0.1 mM non-essential amino acids, 10 mM 4-(2-hydroxyethyl)-1-piperazineethanesulfonic acid (HEPES), 100 U/mL penicillin and 100 μg/mL streptomycin or Dulbecco's Modified Eagle Medium (DMEM) with 10% heat-inactivated fetal bovine serum (FBS), 2 mM glutamine, 100 U/mL penicillin and 100 μg/mL streptomycin. C57BL/6 wildtype (WT), C57BL/6 $A_{2A}R^{-/-}$ and C57BL/6 human-Her2 (hHer2) transgenic mice bred in the Peter MacCallum Cancer Centre animal facility were used in this study. NOD.Cg-Prkdc scid IL2rg (NSG) mice were either bred at the Peter MacCallum Cancer Centre or sourced externally from Australian BioResources (Moss Vale, New South Wales). Mice were used in experiments between the ages of 6 to 16 weeks and housed during that time in PC2 pathogen-free conditions. All experiments were approved prior by the Animal Experimentation Ethics Committee (AEC) at the Peter MacCallum Cancer Centre.

### Reagents, drugs and cytokines
SCH58261 ($A_{2A}R$ antagonist), DPCPX ($A_1R$ antagonist) and adenosine analog 5'-(N-ethylcarboxamido) adenosine (NECA) were purchased from Abcam. Adenosine was purchased from NovaChem. For cell stimulation, anti-CD3 (clone 145-2C11) and anti-CD28 (clone 37.51) antibodies were purchased from BD Pharmingen and anti-myc tag (clone 2276) from Cell signaling Technology. Human anti-CD3 (OKT3) was obtained from Thermo Fisher Scientific. Human IL-2 was obtained from the National Institutes of Health (NIH, Maryland, USA) or purchased from Peprotech, while mouse IL-7 and IL-15 were also obtained from Peprotech. The non homologous end joining (NHEJ) inhibitor, M3814 was purchased from SelleckChem.

### Generation of murine CAR T cells
Murine adenosine $A_1R$ and $A_3R$ cDNA purchased from GenScript (Piscataway, New Jersey, USA) was cloned into the murine stem cell virus (MSCV) vector and linked to either a fluorescent mCherry marker or truncated human nerve growth factor receptor (NGFR) by an IRES sequence. GP+E86 packaging cell lines that generate both anti-HER2 CAR and $A_1R$/$A_3R$ retrovirus in the supernatant was generated as previously described[9,86]. The second-generation anti-Her2 CAR construct is comprised of an extracellular Scfv specific for human Her2, an extracellular CD8 hinge region, a CD28 transmembrane domain and an intracellular CD3ζ domain. Supernatants from these cells were used to transduce primary mouse T cells as previously described[9,21] and following transduction, CAR T cells were maintained in supplemented RPMI media with IL-7 (200 pg/mL) and IL-15 (10 ng/mL).

### Generation of human CAR T cells
Second-generation CAR retroviral or lentiviral constructs comprising of extracellular Scfv targeting human Lewis Y, HER2 or ROR1 antigens,

an extracellular CD8 hinge region, human CD28 or 41BB transmembrane domains and an intracellular human CD3ζ domain were utilized. Human adenosine $A_1R$ cDNA purchased from GenScript (Piscataway, New Jersey, USA) was cloned into the pSAMEN vector and linked to the truncated human nerve growth factor receptor (NGFR) by an IRES sequence. PG13 packaging cell lines that generate both human anti-Lewis Y CAR and human $A_1R$ retrovirus in the supernatant was generated as previously described[86]. Supernatants from these cells were used to transduce primary activated human T cells as previously described[9,21,87]. Cells were then cultured in IL-2 containing media (600 IU/mL).

Fourth generation lentiviral packaging plasmids (pCMV-VSV-G, pMDLg/pRRE, pRSV-Rev) and transfer plasmid vectors encoding the second-generation CAR constructs were purchased from GenScript and transfected into HEK293T cells. Supernatant was collected every 24 h for the following 3 days pooled and centrifuged with Lenti-X-Concentrator (Takara Bio) to concentrate the lentivirus. Lentivirus was used then used to transduce human T cells previously activated with OKT3 (30 ng/mL) for 48 h by adding virus directly to cell cultures at a functional MOI of 0.5-1.0 in 1:400 concentration of Lentiboost (Sirion Biotech). Cells were then cultured in IL-2 containing media (600 IU/mL). We acknowledge the Centre Of Excellence in Cellular Immunotherapy (Victoria, Australia) for the use of their lentivirus backbone plasmid.

### qRT-PCR to quantify adenosine receptor expression

RNA was isolated from T lymphocytes using the Qiagen miRNA Easy Mini Kit per the manufacturer's instructions. cDNA was generated and qPCR analysis of mouse or human $A_1R/A_3R$ against the housekeeping genes RPL32 or GAPDH were determined as previously described[10,88].

### cAMP detection assay

The LANCE ultra cAMP kit (PerkinElmer, catalog no. TRF0262) was used for this assay. 4x NECA or 4x eADO (50-(N-Ethylcarboxamido) adenosine, Abcam) was serially half-log diluted in stimulation buffer (HBSS containing 5 mM HEPES, 0.1% BSA, 25 μM rolipram, pH adjusted to 7.4). A standard curve was generated with a serial half log dilution of included cAMP standard from the kit. Subsequently, 5 uL of 2x NECA and 10 uL 1x cAMP standard were added to wells prior to addition of T cells. CAR T cells were harvested from culture, washed twice and resuspended to $1 \times 10^6$ cells/mL in stimulation buffer. CAR T cells were pre-incubated with 1μM of Forskolin (Abcam) for a total of 30 min before transfer of 5 μL of cell suspension to each well of a 384-well plate containing 2x NECA, thus adjusting the final volume to 10 μL/well. Cells were then incubated with drugs for another 30 min at room temperature. 4x Eu-cAMP tracer and 4x U-light anti-cAMP antibody were prepared in provided detection buffer at 1:50 and 1:150 dilution respectively, and 5 μL of Eu-cAMP mix and 5 μL U-light mix were then added in that order to each well of standard curve and cell suspension. Plates were then incubated for a minimum of 1 hour at room temperature and protected from light before storing at 4 °C prior to measurement on an EnVision Multimode plate reader system (Perkin Elmer). The TR-FRET signal (665 nm) was plotted against the concentration of cAMP to generate a non-linear standard curve. Experimental cAMP concentrations were then interpolated from the standard curve.

### CRISPR/Cas9 editing and Homology Directed Repair (HDR) Knock-in in CAR T cells

CRISPR/Cas9 editing of murine and human CAR T cells was performed as previously described[9]. 270 pmoles sgRNA (Synthego) and 37 pmoles recombinant Cas9 were combined and incubated for 10 min at room temperature to generate Cas9/sgRNA RNP, in which $5 \times 10^6$ activated human PBMCs per large cuvette (Lonza) were resuspended to 20 μl P3 buffer (Lonza) or X-VIVO (Lonza) media. Cells were then electroporated with a 4D-Nucleofector (Lonza) using

pulse code E0115. Pre-warmed media was added to cells and cells were rested for 10 min prior to the transfer to 96 well round bottom plates pre-loaded with 10k MOI of AAV6 purchased from Packgene Biotech (Houston, USA) encoding human NGFR or $A_1R$ and 2uM of M3814 drug (NHEJ inhibitor) and incubated for a further 4 h. Post incubation, activated human T cells were transduced with either retrovirus or lentivirus. sgRNA sequences used were as follows: *NR4A2 sgRNA: gccugaacacaaggcauggc, IRF8 sgRNA: gcguaaccucgucuuccaag, ADORA2 sgRNA: cuacuuuguggugucacugg*

### In vitro serial antigen and re-stimulation assay

Tumor cell targets were co-cultured with CAR T cells at a 1:1 effector to target ratio for 24 h. For serial antigen stimulation, additional tumors were added at 48 and 72 h at the same E:T ratio with fresh media. At each step, cells and supernatant were harvested for analysis by flow cytometry and cytometric bead array (CBA) according to the manufacturer's instructions to quantify cytokine production. CAR T cells were cocultured with tumors for 72 h, before washing and reseeding with fresh media and IL-2 (600 U/mL). After resting the cells for 4 days, CAR T cells were restimulated with more tumors. This process was repeated twice after the first stimulation for up to a total of 2 additional re-stimulations. At each time-point prior to reseeding, supernatant and cells were harvested for analyses by CBA and flow cytometry respectively.

### Incucyte killing and chromium killing assay

Cytotoxic capacity of tumor-specific CAR T cells were assessed in 4-hour $^{51}$Chromium release assays. Tumor targets were washed in serum free media and resuspended in 100 μCi/million cells of chromium and incubated at 37 °C for 1 hour. During incubation, CAR T cells were counted and plated in in V-bottom 96 well plates at highest E:T ratio and a half serial dilution down to the lowest E:T ratio as specified. NECA was added to cells at indicated concentrations. CAR T cells were cocultured with $^{51}$Cr labelled tumor cells. A minimum and maximum killing control was setup whereby tumor cells are plated with media alone or lysed with 100% sodium dodecyl sulphate (SDS) respectively. After the incubation period, supernatant was measured for radioactivity in an automatic gamma counter Wallac Wizard 1470 (Amersham Australia, now General Electricity Healthcare). Killing was determined as a percentage of maximum killing minus minimum background readings. For Incucyte killing assays (Sartorius), tumor cells were transduced to express a fluorescent marker (mCherry or GFP) and seeded out in a 384 well plate. CAR T cells were then cocultured with tumors, with the addition of the manufacturer provided Caspase 3/7 dye and killing quantified by the provided image analyses software.

### Flow cytometry and cell sorting

Cells were stained with fluorochrome-conjugated antibody cocktails and incubated for 30 min in the dark at 4°C. Where required with mouse cells, an Fc receptor block (2.4G2 diluted 1:50 from hybridoma supernatant in FACS buffer) was incubated with cells for 10 min prior antibody staining to prevent non-specific binding of antibodies. For intracellular staining, cells were fixed and permeabilized using the eBioscience™ FoxP3 / Transcription Factor Staining Buffer Set (Thermofisher) according to the manufacturer's instructions. Samples were quantified using counting beads (Beckman Coulter; 20 μL per sample) using the following formula: number of beads per sample/bead events x cell events of interest. All flow cytometry analyses were performed on either the BD LSRFortessa or BD FACSymphony (BD Biosciences) and data was analyzed using Flowjo (TreeStar). Cells were sorted using a BD FACSAria Fusion.

### Treatment of mice with CAR T cells

Female C57BL/6 human Her2 transgenic mice were injected in the fourth mammary fat pad with 2x10^5 E0771-Her2 cells. After tumor

establishment 6–7 days after injection, mice were given 4 Gy total-body irradiation (TBI) as a preconditioning regiment prior to the intravenous (i.v) administration of two doses of $1 \times 10^7$ anti-Her2 CAR T cells on subsequent days. Mice were also intraperitoneally (i.p) injected with 50,000 IU of hIL-2 on days 0-4 post CAR T cell therapy. Measurements were made every 2–3 days of tumor size, with the endpoint set to 100 mm², which is lower than the maximum tumor size/burden permitted by the ethics committee of 150mm². For human adoptive transfer experiments of anti-Lewis Y CAR T cells, NSG mice were injected subcutaneously with $5 \times 10^6$ OVCAR-3 ovarian cancer cells. Once tumors were established (-20 mm²) mice were preconditioned with 1 Gy TBI before i.v injection of up to $1 \times 10^7$ anti-Lewis Y CAR T cells on two consecutive days. NSG mice were also treated with 50,000 IU of hIL-2 for days 0–4 post adoptive transfer. NSG mice treated with human anti-Her2 CAR-T cells were injected with $1.25 \times 10^6$ MDA-MB231 breast cancer cells in the mammary fat pad. After establishment of tumors the mice were irradiated with 1 Gy TBI prior to receiving a single dose of $5 \times 10^6$ anti-Her2 CAR T cells intravenously, followed by the same regimen of hIL-2.

## Analysis of immune subsets in tumor, spleen, draining lymph nodes and blood

To collect blood from mice, submandibular or retroorbital bleed procedures were performed into tubes containing EDTA. Red blood cells from blood and spleen samples were lysed with ACK lysis buffers prior to staining for flow cytometry. Tumors were mechanically digested and enzymatically digested with DMEM supplemented with 1 mg/mL of collagenase type IV (Sigma-Aldrich) and 0.02 mg/mL DNAase (Sigma-Aldrich). Tumors were incubated in this media for 30 min with constant shaking at 37 °C. Tumor single cell suspensions were then filtered twice through (70 µm filters) before extracellular or intracellular flow cytometry staining. Spleens and dLN were mechanically digested and filtered through 70 µm filters prior to staining. For stimulation of intratumoral CAR T cells to assess cytokine secretion capacity, single cell suspensions were stimulated with PMA (5 ng/mL) and ionomycin (1 µg/mL; Sigma-Aldrich) with GolgiPlug and GolgiStop (BD Biosciences) for 3–4 h prior to downstream flow cytometry staining. The resultant cell suspension was then stained for analysis by flow cytometry.

## Gene expression and network analysis

Following manufacturer's instructions, preparation of RNA-seq libraries from mRNA was done using the Quant-seq 3' mRNA-seq Library Prep kit for Illumina (Lexogen) as per manufacturer's instructions. NextSeq (Illumina, Inc., San Diego, CA) sequencing was then performed to generate single-end, 75–100 bp RNA-seq short reads and CASAVA 1.8.2 was used for base calling. Quality control was assessed using FastQC v0.11.6 and RNA-SeQC v1.1.8[89]. Cutadapt v2.1 was used to remove random primer bias and remove 3' end poly-A-tail derived reads. Sequence alignment against the mouse reference genome mm10 or the human genome hg19/hg38 was performed using HISAT2. Finally, featureCounts from the Rsubread software package 2.10.5 was used to quantify the raw reads with genes defined from the respective Ensembl releases[90]. Gene counts were normalized using the TMM (trimmed means of M-values) method and converted and filtered on log2 counts per million (CPM) using the EdgeR package[91,92]. The quasi-likelihood F test statistical test method based on the generalized linear model (glm) framework from EdgeR was used for differential gene expression comparisons. Adjusted p values were computed using the Benjamini-Hochberg method. Principal component analysis (PCA) was performed generated based on the topmost variable genes. Differentially expressed genes (DEGs) were classified as significant based on a false discovery rate (FDR) cutoff of less than 0.05. For heatmaps, the pheatmap R package was used to plot row mean centered and scaled normalized log2(CPM + 0.5) values. Genes columns or rows were sorted by hierarchical clustering using Euclidean distance and average-linkage.

Unbiased gene set enrichment analysis was performed using fgsea package on differential expressed genes pre-ranked by fold change with 1000 permutations (nominal P-value cutoff <0.05)[93]. Reference gene sets were obtained from the MsigDB library for Hallmarks, KEGG, single-cell RNA sequencing derived T cell clusters in patients[55].

Weighted gene co-expression network analysis (WGCNA) was performed to obtain a holistic understanding of gene expression patterns between samples[94]. Briefly, normalized counts from sorted CD8 + T cell samples were correlated and transformed into a topological overlap matrix. A soft thresholding power of $\beta = 14$ was employed to achieve scale-free topology. Using a minimum module size = 5, merge cut height = 0.1, co-expressing gene modules were partitioned from the data. Module eigengene values were used as an estimate of overall gene expression within that module which is based on the first principal component. Gene hub score (normalized to 0-1), measured by intramodular connectivity (k-within), was calculated by the sum of all correlation strengths for any given gene against all other genes within its module. Gene expression networks for each module were visualized using Cytoscape v3.6.0.

## ATAC-Seq data analysis

Sequencing files for ATAC-seq experiments were demultiplexed using Bcl2fastq (v2.20) to generate Fastq files. Next QC of files were performed using FASTQC (v0.11.5). Adaptor trimming of paired-end reads was performed with NGmerge (v0.3) where required[95]. Alignment of reads to either the reference human (hg38) genome was performed using Bowtie2 (v2.3.3). The resulting SAM files were converted to BAM files using Samtools (v1.4.1) using the view command, which were subsequently sorted and indexed, with potential PCR duplicates marked with Samtools markdup. Peak calling was performed with either MACS2 (v2.1.1) or Genrich (v0.6.0) packages. Annotation of ATAC-Seq peaks to proximal genes was performed using either annotatePeaks.pl (Homer, v4.11) or the annotatePeak function from ChIPseeker R package (v1.8.6). BAM files were converted into BigWig files using the bamCoverage function (Deeptools, v3.5.0). BigWig files were then imported into Integrative Genomics Viewer (IGV, v2.7.0) for visualization of specific loci. To generate IgV style track plots from BigWig files, the package trackplot was used[96]. The HOMER make-TagDirectory command was used to generate tag directories, and the findPeaks command was used to identify peaks, with the control tag directory set to respective control groups. Motif discovery using the findMotifsGenome tool and default settings identified de novo motifs from peaks identified.

## Statistical analysis

Statistical analyses were performed using GraphPad Prism. Analyses performed include paired or unpaired Student's t test to compare two data sets, one-way ANOVA to analyze multiple data sets across a single time point and two-way ANOVA when analyzing multiple sets of data across time.

## Reporting summary

Further information on research design is available in the Nature Portfolio Reporting Summary linked to this article.

## Data availability

The RNA and ATAC sequencing data that support the findings of this study have been deposited in GEO NCBI under the accession code GSE284619, GSE284618, GSE284616, GSE284615, GSE284614. Hallmarks, Gene Ontology (GO), Pathway Interaction Database (PID), KEGG and Immunological signature gene sets utilized can be accessed via https://www.gsea-msigdb.org/gsea/msigdb/index.jsp. Source data are provided with this paper as a Source Data file. Reference databases

used are hg38 (https://www.ncbi.nlm.nih.gov/datasets/genome/GCF_000001405.26/), mm10 (Mus musculus genome assembly GRCm38−NCBI−NLM (nih.gov) and hg19 (https://www.ncbi.nlm.nih.gov/datasets/genome/GCF_000001405.13/). Antibodies and clones used in this paper are provided in the Supplementary Data file. The remaining data are available within the paper, supplementary information, and tables or available upon request from the authors. Source data are provided with this paper.

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

## Acknowledgements

This work was funded by a Program Grant from the National Health and Medical Research Council (NHMRC; Grant number 1132373), an NHMRC Project grant (APP1122444), a Peter MacCallum Foundation Grant, a Grant-in-aid (Cancer Council Victoria; Grant number 836379), and a Synthego Genome Engineer Innovation Grant. P. A. Beavis was supported by a National Breast Cancer Foundation Fellowship (ID# ECF-17-005) and is currently supported by a Victorian Cancer agency fellowship (MCRF20011) and is a CRI Lloyd J Old STAR (CRI5578). The Guimaraes and Beavis Laboratories were funded by a US Department of Defense – Breast Cancer Research Program – Breakthrough Award Level 1 (#BC200025) P. K. Darcy was supported by an NHMRC Senior Research Fellowship (APP1136680). I.A. Parish is supported by a Victorian Cancer agency fellowship (MCRF21019).

## Author contributions

K.S., A.X.Y.C., P.K.D., and P.A.B. designed the experiments, developed the methodology, interpreted the data, and wrote the paper. I.G.H. and K.M.Y. also developed the methodology. K.S., A.X.Y.C., T.C., J.T., K.M.Y., I.M., P.A.D., S.W., I.G.H., P.K.D., and P.A.B. performed the experiments and acquired the data. J.D.A., M.J.E., O.H., C.S., L.G., M.A.H., D.M., F.S.G., D.N., Y.H., M.N.M., E.B.D., C.W.C., K.L.T., J.D.C., J.L., JY.L., E.V.P., S.M., A.B., J.W., I.A.P., C.M., G.D.S., L.K. provided the technical assistance and advice on the data analysis and interpretation. K.S., P.K.D., and P.A.B. supervised the study and were responsible for coordination and strategy.

## Competing interests

P.A.B. declares the following conflicts; Research funding: Gilead, Bristol Myers Squibb. P.K.D. declares the following conflicts: research funding from Myeloid Therapeutics, Prescient Therapeutics and Bristol-Myers-Squibb. K.S. declares the following conflicts: Research funding: Prescient Therapeutics. I.A.P. declares the following conflicts; Research funding: AstraZeneca, Bristol Myers Squibb. A.B. declares the following conflicts: Founder of INSiGENe Pty Ltd, which is related to this work. Co-founder and director of Respiradigm Pty Ltd that is unrelated to this work. All other authors declare no conflict of interest.

## Additional information

[1]Cancer Immunology Program, Peter MacCallum Cancer Centre, Melbourne, VIC, Australia. [2]Sir Peter MacCallum Department of Oncology, The University of Melbourne, Parkville, VIC, Australia. [3]University of Western Australia, Perth, WA, Australia. [4]Telethon Kids Institute, Perth, WA, Australia. [5]Frazer Institute, Faculty of Medicine, The University of Queensland, Woolloongabba, QLD, Australia. [6]Asthma and Airway Disease Research Center, The University of Arizona, Tucson, AZ, USA. [7]Drug Discovery Biology, Monash Institute of Pharmaceutical Sciences, Monash University, Parkville, VIC, Australia. [8]These authors contributed equally: Kevin Sek, Amanda X. Y. Chen. [14]These authors jointly supervised this work: Phillip K. Darcy, Paul A. Beavis. ✉e-mail: kevin.sek@petermac.org; phil.darcy@petermac.org; paul.beavis@petermac.org

