## [Transparent Peer Review file · Nature Communications]

Tumor-site directed A1R expression enhances CAR T cell function and improves efficacy against solid tumors

Corresponding Author: Professor Phillip Darcy

Version 0:

Reviewer comments:

Reviewer #1

(Remarks to the Author)

Sek et al have submitted a manuscript where they investigate the role and potential of A1AR signaling in CAR T cell function in relationship to suppressive A2AR receptors. They show that while A1AR signaling may enhance cytokine production capabilities in vitro, on the long run this positive effect is countered by A2AR signaling which seem to drive T cell differentiation away from SCM T cells. Thus simultaneous knock of of A2AR and A1AR expression circumvents this effects and results in bettered outcomes in vivo, due to IRF8 signaling. The manuscript is well written and mostly sound. The actual novelty comes from the use of A1AR, preferably in the absence of A2AR, the rest of the techniques being known. I have the following concerns:

- 1) Overall I do not agree with the notion that A1AR expression in the absence of A2AR KO would be worth then CAR in vivo. This is not backed up by data and should be corrected to non-different.
- 2) It is surprising that while A2AR has a known suppressive effect on CAR T cells. Knock out thereof does not seem to lead to expected outcome. This needs further elaboration
- 3) The amount of TSCM after CAR T cell generation seem quite high with 20+%, how is this protocol compared to regular expansion protocols typically at best in the single digit range?
- 4) A major concern is the focus on a single model to demonstrate utility: OVCAR, which is very CAR permissive. Especially in the context of comment 1). More evidence in different models and ideally with different CAR is needed to allow generalization of the conclusions. Otherwise this is a one model thing.
- 5) Increased cytokine secretion is not universally good and can be actually quite detrimental in terms of safety. What is the contribution of cytokines to the observed effects? The SCM relationship would typically be more driven towards T cell fitness.
- 6) For each and every figure and subfigure the amount of independent repeats and replicates as well as donors must be clearly stated
- 7) How was cell line identity ascertained.

Reviewer #2

(Remarks to the Author)

In this manuscript, Sek et al. ectopically expressed A1R to overcome adenosine-A2AR-mediated T cell exhaustion. The authors first tested A1R overexpression and found that this strategy did not improve overall antitumor immunity due to impaired T-cell persistence. They then investigated the effect of activation-dependent A1R expression by knocking in the gene under the NR4A2 promoter. The authors showed that this strategy can overcome the problem of constitutive A1R expression. I have the following specific comments.

1. In Figure 1B, the authors analyzed cAMP levels only under artificial experimental conditions: exogenous addition of adenosine to forskolin-stimulated T cells). Please show the data that A1R overexpression can reduce intracellular cAMP levels in a more physiological setting (e.g., antigen-stimulated CAR-T cells without adenosine treatment).
2. Please show cell-surface expression of the A1 receptor in control and overexpressing T cells analyzed by flow cytometry.

Several antibodies are commercially available.

3. In the previous paper by the same group (Giuffrida et al. Nat Commun 2021), the authors showed that genetic ablation of ADORA2A enhances both effector functions and persistence of CAR-T cells in mouse tumor models. The data presented in Figure 2 of this manuscript appear to be inconsistent with these findings. How do the authors reconcile this discrepancy? If it is due to a different experimental protocol, the authors should confirm whether A2AR-knockout CAR-T cells show similar profiles in the present protocol (accelerated differentiation and poor persistence). Another possibility is that A1R overexpression may affect T cell function by other mechanisms in addition to suppression of A2AR-mediated signaling.

4. In Figure 3, please show the knock-in efficiency of multiple experiments.

5. How long does A2AR expression persist after antigen stimulation? Supp. Figure 5 only shows its upregulation 16 hours after stimulation. In addition, is it constitutively expressed in the tumor-infiltrating CAR-T cells in vivo because they are chronically exposed to the antigen?

6. A1R upregulation under the control of the physiological NR4A2 promoter will be lower than that driven by a conventional retroviral promoter even after antigen stimulation, which may be a reason for the attenuated terminal differentiation and improved persistence of CAR-T cells. The authors should directly compare A1 receptor expression at the protein level.

7. In Figure 4a-c, the authors claim that tumor progression was better controlled by NR4A2/hA1R CAR-T cells. However, tumor growth was also suppressed in most of the mice treated with control CAR-T cells and CAR-T cells with NR4A2 knockout (red and blue lines). In addition, there was no significant difference in the CD8+ T cell counts between control and NR4A2/hA1R CAR-T cells.

The authors should optimize a protocol to precisely determine the antitumor effects of NR4A2/hA1R CAR-T cells (e.g., reducing the CAR-T cell dose or infusing CAR-T cells at later time points). In addition, a longer follow-up period would be needed to evaluate the persistence of CAR-T cells.

8. The authors should confirm that IRF8 expression is increased by A1R expression and diminished upon CRISPR/Cas9 knockout at the protein level. What was the efficiency of the knockout as assessed at the genomic level?

9. The IRF8-knockout data are not convincing. While they did not affect cytokine secretion by CAR-T cells in Supp. Figure 8C, it was significantly suppressed in Figure 5I. The authors should provide a more convincing explanation for these different results. Also, CD62L+ CD69- cells cannot be defined as memory T cells.

10. Most critically, what is the overall benefit of the strategy of expressing A1R under the NR4A2 promoter compared to the simple knockout of the A2A receptor presented in the previous paper? Since CRISPR/Cas9-mediated gene knock-in is a more complicated procedure, it will not be practically useful unless it has some functional advantage over the knockout of A2A receptor. It will be necessary to directly compare the in vivo efficacy, persistence and representative phenotypes of CAR-T cells between control, A1 receptor overexpression, NR4A2-A1R expression, and A2A receptor KO CAR-T cells.

Reviewer #3

(Remarks to the Author)

I really liked the premise for your paper based on your earlier work. And a really cool advance to limit A1R expression to T cells traversing the TME.

Reviewer #4

(Remarks to the Author)

In this extensive and impressively illustrated study, the authors report that engineering of CAR T cells with A1R enhances their efficacy against solid tumors in vitro and in vivo. This is a highly relevant study that could impact therapy of solid tumors with CAR T cells. The latter has not been as satisfactory as expected: only a small percentage of patients treated with CAR T cells achieve CR. This lack of efficacy is attributable to various barriers, but the authors focus on ADO-mediated immune suppression as the major check point for tumor-directed CAR T cells.

Here, they describe "arming" of CAR T cells against immunosuppressive TME by overexpression of A1R, which attenuates suppressive effects of A2AR, a major ADOR in T cells, by inhibition of adenylate cyclase and cAMP accumulation. The authors show that overexpression of A1R and not of A3R suppressed cAMP production and enhanced IFN γ , TNF and granzyme activities in both murine and human CAR T cells. However, constitutive A1R expression resulted in a loss of CAR T cell persistence and reduction in the T stem cell memory (TSCM) subset. So, the authors opted for transient A1R expression by restricting A1R signaling to the tumor site, where ADO is produced. Restriction of A1R to the tumor site was accomplished by using a novel CRISPR knock-in approach under control of the NR4A2 promoter. This CAR T cell modification drove their differentiation and promoted anti-tumor functionality without having any impact on T cell persistence. Thus, the ability to transiently overexpress A1R at the tumor site led to improved therapeutic efficacy by activation of T cells within the tumor but not the spleen. Overexpression of A1R restricted ADO produced by the tumor and reduced ADO mediated suppression of CAR T cells.

While the experimental design is rational and experiments were rigorously controlled, there are some concerns. These largely consists of a potential transfer of this methodology to the clinic. First, knock-in of A1R gene, while feasible, might alter the signaling balance of ADORs in T cells as well as non-immune cells in an unexpected way. Are there any offtarget effects that the authors observed? Another concern transient effects of relates to A1R site-specific overexpression; how will timing of

adoptive transfers be orchestrated to achieve optimal therapeutic effects? Also, the authors are overly optimistic about therapeutic efficacy of this therapy targeting but a single suppressive pathway. Although the ADO may be the major checkpoint in some HNCs, it is not the only one. Numerous other immunosuppressive pathways exist, and may differ in functional intensity among patients. Is blockade of one pathway be sufficient to save CAR T cells from tumor-induced suppression??.

The manuscript is clearly written, and the data presentation is excellent.

Version 1:

Reviewer comments:

Reviewer #1

(Remarks to the Author)

The authors have in principle addressed my concerns.

Reviewer #2

(Remarks to the Author)

The authors conducted extensive additional experiments. All of my comments were addressed appropriately.

Reviewer #3

(Remarks to the Author)

Revised work has steadily improved the manuscript. All concerns have been dealt with.

Point by Point Response:

Reviewer #1 (Remarks to the Author): with expertise in CAR-T cells

Sek et al have submitted a manuscript where they investigate the role and potential of A1AR signaling in CAR T cell function in relationship to suppressive A2AR receptors. They show that while A1AR signaling may enhance cytokine production capabilities in vitro, on the long run this positive effect is countered by A2AR signaling which seem to drive T cell differentiation away from SCM T cells. Thus simultaneous knock of of A2AR and A1AR expression circumvents this effects and results in bettered outcomes in vivo, due to IRF8 signaling. The manuscript is well written and mostly sound. The actual novelty comes from the use of A1AR, preferably in the absence of A2AR, the rest of the techniques being known.

Response: We thank the reviewer for their positive critique of our paper. We believe that the reviewer may also be referring to our data where A₁R was knocked in to the NR4A2 locus rather than a ‘simultaneous knock-out’ of A_{2A}R. This was done in order to achieve tumor-site specific expression of A₁R.

I have the following concerns:

1) Overall I do not agree with the notion that A1AR expression in the absence of A2AR KO would be worth then CAR in vivo. This is not backed up by data and should be corrected to non-different.

Response: Our original manuscript did not report data with A_{2A}R knock-out but rather NR4A2 knockout. We apologise for this confusion, and we have sought to clarify this further in the text and in **Supp Figure 5A**.

See additional text: *“Furthermore, it is important to note that this approach simultaneously inserts the A₁R gene while knocking out NR4A2 (Supp Fig 5A).”* in lines 491-493.

2) It is surprising that while A2AR has a known suppressive effect on CAR T cells. Knock out thereof does not seem to lead to expected outcome. This needs further elaboratio

Response: Please see response to Reviewer 1 point 1) above.

3) The amount of TSCM after CAR T cell generation seem quite high with 20+%, how is this protocol compared to regular expansion protocols typically at best in the single digit range?

Response: Our expansion protocol is based upon previously published work from our labs and others, and our results are consistent with these studies (Fraiatta *et al.* 2018 *Nature Medicine*, Chan *et al.* 2024, *Nature*). Our TSCM gating strategy based off these two publications is shown in **Supp Figure 3B** (Mouse) and **Supp Figure 3C-D** (Human).

4) A major concern is the focus on a single model to demonstrate utility: OVCAR, which is very CAR permissive. Especially in the context of comment 1). More evidence in different models and ideally with different CAR is needed to allow generalization of the conclusions. Otherwise this is a one model thing.

Response: We agree with the reviewer that data in a second model would strengthen our conclusions. In our revised paper, we evaluate the effect of inducible A₁R expression in an additional *in vivo* tumor model as well as *in vitro* analysis of 3 alternative CAR T cell antigen systems using both retroviral and lentiviral transduction protocols. This additional data has now been included as additional panels in **Supp Fig 6D-F** and **Figure 4A**.

- See lines 513-522 in Results section:

*“To evaluate the broad applicability of NR4A2/hA₁R to enhance T cell adoptive therapy, we also applied this engineering approach to CARs targeting the HER2 and ROR1 antigens and in the context of CARs with either a CD28 or 4-1BB signaling domain. Consistent with our results obtained with anti-Lewis Y CAR T cells, NR4A2/A₁R engineered anti-HER2 CAR T cells exhibited significantly enhanced cytokine production upon coculture with HER2⁺ MCF-7 breast tumors (Fig 3I, Supp Fig 6C). Regardless of costimulatory domains expressed, HER2-CARs with CD28 or 41BB domains secreted enhanced cytokine production against MDA-MB231 breast tumors (Supp Fig 6D-E). Similarly, NR4A2/hA₁R engineering enhanced the cytokine production of anti-ROR1 CAR T cells when cocultured with MDA-MB231 tumors *in vitro* (Supp Fig 6F).”*

- Enhanced *in vivo* efficacy and survival in a HER2 breast cancer model by A₁R edited anti-HER2 CAR T cells (**Figure 4A**). This data is referred to in lines 545-547 of the text:

*“To determine the *in vivo* efficacy of NR4A2/hA₁R engineered CAR T cells, 5x10⁶ anti-HER2 CAR T cells or 15x10⁶ anti-Lewis Y CAR T cells were adoptively transferred to treat MDA-MB231 breast (Fig 4A) or OVCAR-3 ovarian (Fig 4B) tumor bearing mice respectively.”*

5) Increased cytokine secretion is not universally good and can be actually quite detrimental in terms of safety. What is the contribution of cytokines to the observed effects? The SCM relationship would typically be more driven towards T cell fitness.

Response: The reviewer has highlighted a recent important finding in the CAR-T field which has found that anti-tumor cytokines are potentially dispensable in liquid tumors, however they are critical for an effective response in the context of solid tumors as demonstrated by Larson *et al.* 2021, *Nature*. (PMID 35418687). We agree with the reviewer that cytokine secretion by CAR T cells can be detrimental in certain contexts, but this is more commonly associated with hematological cancers (e.g. PMID 37296093). Furthermore, we have also previously shown this to be the case following IFN γ blockade in our solid tumor models following adoptive transfer of CAR T cells as used in the current study (Beavis *et al.* *JCI* 2017; Chan *et al.* *Nature*, 2024). Assessment of the requirement for enhanced cytokine production in the anti-tumor efficacy for NR4A2/A₁R engineered CAR T cells was beyond the scope of our study but we have provided additional discussion on this point on line 91-94 of the revised manuscript:

“In this work, we show that expression of A₁R but not A₃R suppresses cAMP production in CAR T cells and enhances both murine and human CAR T cell function by promoting their production of IFN γ and TNF, cytokines that have been demonstrated to be critical for anti-tumor efficacy of CAR T cells in solid tumors [19, 29, 30].”

6) For each and every figure and subfigure the amount of independent repeats and replicates as well as donors must be clearly stated.

Response: This information has now been included in the revised manuscript

7) How was cell line identity ascertained.

Response: We utilize short tandem repeat (STR) analyses to confirm the identity of our cell lines and used within 10 passages of a master stock to ensure their accuracy.

Reviewer #2 (Remarks to the Author): with expertise in CAR-T cells

In this manuscript, Sek et al. ectopically expressed A1R to overcome adenosine-A2AR-mediated T cell exhaustion. The authors first tested A1R overexpression and found that this strategy did not improve overall antitumor immunity due to impaired T-cell persistence. They then investigated the effect of activation-dependent A1R expression by knocking in the gene under the NR4A2 promoter. The authors showed that this strategy can overcome the problem of constitutive A1R expression. I have the following specific comments.

1. In Figure 1B, the authors analyzed cAMP levels only under artificial experimental conditions: exogenous addition of adenosine to forskolin-stimulated T cells). Please show the data that A1R overexpression can reduce intracellular cAMP levels in a more physiological setting (e.g., antigen-stimulated CAR-T cells without adenosine treatment).

Response: We thank the reviewer for this question - it raises some interesting points around 1. the activity of forskolin at adenylate cyclase and 2. some sensitivity aspects of the assay.

1. Forskolin is a small molecule activator of adenylate cyclase that acts in synergy with GalphaS proteins via a structurally distinct site (PMID 38351373). Forskolin acts to potentiate the activity of GalphaS (in this case working via adenosine-mediated activation of the A_{2A}R) and can increase the rate of conversion of ATP to cAMP. The use of a sub-maximal concentration of forskolin will 'prime' the adenylate cyclase for activation by GalphaS post-A_{2A}R activation. Together, in lieu of high A₁R constitutive activity, the addition of exogenous ADO and concomitant A₁R and A_{2A}R activation and forskolin-priming allows us to demonstrate the modulatory effect of the A₁R on T cell cAMP.

2. The design of the cAMP kit is a competition assay between cAMP in cell lysates and a cAMP probe, which means the response has an upper and lower limit based on the amount of antibody and probe used in the assay (recommended to not alter reagent ratios) and that the cAMP in lysates needs to be in the dynamic range of the system (0.5-15nM), which is why we use a forskolin-priming approach and ADO to raise the level of cAMP into that range. Low local levels of cAMP still have big impacts on biological outcomes, but may well be below detectable limits with the technologies available. Hence, we feel our assay design gives a correct appraisal of the ADO-mediated cAMP dynamics and the overall effect of A₁R overexpression and activation.

We have further included **Reviewer Figure 1** below for the optimisation of the cAMP assay. We show that the priming concentration of forskolin (1µM) was selected so as to not drive cAMP in primary T cells on its own, and that this priming was essential for achieving the dynamic range in this assay when NECA or ADO was added.

Thus, we are confident that our data indicates that A₁R expression can inhibit A_{2A}R mediated cAMP signaling. We also note that our data indicates that A₁R knock-in induces a distinct transcriptional program relative to A_{2A}R KO (please refer to response to Point 3 Reviewer 2), suggesting that modulation of cAMP signaling is only one component by which A₁R expression enhances CAR T cell effector function.

Reviewer Figure 1. Optimization of Lance Ultra cAMP assay for detecting intracellular cAMP in CAR T cells. Control CAR T Cells at days 7-9 post transduction were incubated with Forskolin (FSK) for 30 minutes before internal cell lysates were used to detect cAMP. A. TR-FRET lance output based on the standard curve of known concentrations of cAMP B. TR-FRET lance output of 1,000-20,000 CAR T cell numbers incubated for 30 minutes with increasing concentrations of FSK. C. cAMP readout of 5000 CAR T cells after incubation with FSK for 30 minutes. D. Control CAR T cells were incubated with/without FSK (1 μ M) for 30 minutes, before NECA was added at varying concentrations for another 30 minutes. Triplicate means \pm SD from n=1 experiment ****p<0.0001, *p<0.05, One-way ANOVA

2. Please show cell-surface expression of the A₁ receptor in control and overexpressing T cells analyzed by flow cytometry. Several antibodies are commercially available.

Response: In our experience, antibodies against adenosine receptors lack specificity and sensitivity (which is why we previously generated an A_{2A}R reporter mouse; Todd *et al.* 2023 Nature Communications). Although we were unable to evaluate potential endogenous A₁R

protein expression, our mRNA data indicate that control T cells express almost undetectable levels of A₁R mRNA, a result that has previously been reported by other groups (PMID 11992407).

However, to address this question, we have labelled the hA₁R with a MYC tag while the anti-LeY CAR is labelled with a FLAG tag. Utilizing both tags, we demonstrate constitutive co-expression of A₁R and CAR proteins on the cell surface of lentivirally transduced primary human T cells. Furthermore, CD69 upregulation by constitutive A₁R expression was consistent with our untagged A₁R construct indicating a functional A₁R phenotype. This data has been included as a panel in our modified paper **Supp Fig 2C-D** and referred to in the following text on lines 374-378. The full engineered myc-tag hA₁R protein sequence is provided in **Supp Table 1**.

“Due to the lack of reliable flow cytometry antibodies for the hA₁R, a short Myc-tag sequence was engineered onto the extracellular region of the wild-type hA₁R to confirm protein expression on the cell surface (Supp Fig 2C, Supp Table 1). Co-expression of Flag-tag CAR and Myc-tag hA₁R was determined by flow cytometry, confirming successful cell surface expression of A₁R on anti-Lewis Y CAR T cells (Supp Fig 2C-D)”.

3. In the previous paper by the same group (Giuffrida et al. Nat Commun 2021), the authors showed that genetic ablation of ADORA2A enhances both effector functions and persistence of CAR-T cells in mouse tumor models. The data presented in Figure 2 of this manuscript appear to be inconsistent with these findings. How do the authors reconcile this discrepancy? If it is due to a different experimental protocol, the authors should confirm whether A_{2A}R-knockout CAR-T cells show similar profiles in the present protocol (accelerated differentiation and poor persistence). Another possibility is that A₁R overexpression may affect T cell function by other mechanisms in addition to suppression of A_{2A}R-mediated signaling.

Response: We thank the reviewer for their insightful comments and we believe that the reviewer is correct that A₁R is affecting T cell function through mechanisms other than suppression of A_{2A}R mediated signaling. Therefore, the data shown in Figure 2 should not be considered a discrepancy in relation to previous experiments with A_{2A}R KO CAR T cells, which exhibit a distinct phenotype based on the different mechanisms of the two engineering strategies.

Indeed, the reviewer has raised important questions on the overlap in signaling pathways between the A_{2A}R and A₁R, and the potential for synergy between A_{2A}R deletion and A₁R tumor-site specific expression.

To address this we have therefore performed an *in vitro* assessment of function and transcriptional activity of A₁R knock-in vs A_{2A}R knockout cells in the presence of the adenosine analogue NECA or A₁R inhibitor DPCPX to demonstrate the differences between the two approaches. Furthermore, we have combined A_{2A}RKO and NR4A2/A₁R, demonstrating potential synergy *in vitro* (**Reviewer Figure 2**).

Reviewer Figure 2. A₁R expression and A_{2A}R deletion in CAR T cells differentially regulate cytokine expression. Cytokine production in engineered CAR T cells that were serially stimulated with OVCAR tumors for 72 hours in the presence or absence of pan-adenosine receptor agonist NECA (10 μ M) normalised against NR4A2 CAR-T cells without NECA control. **** p <0.0001, *** p <0.001, ** p <0.01, * p <0.05. (a, b) two-way ANOVA. Data represented as mean \pm SD from 2 independent donors.

We have included the new data in **Supp Figure 8**, showing that NR4A2/A₁R expression drives significantly enhanced cytokine production of IFN γ and TNF upon CAR T cell activation, which was not observed with A_{2A}R deletion (**Supp Fig 8A**). Furthermore, we demonstrated that, consistent with our previous studies (Giuffrida, Sek *et al.* 2021, Beavis *et al.* 2017), a pan-adenosine receptor agonist, NECA, could suppress IFN γ and TNF in control CAR T cells, but that this effect was lost in A_{2A}R KO CAR T cells showing that suppression of cytokines is A_{2A}R mediated (**Supp Fig 8A**). Interestingly, while NECA suppressed IFN γ in NR4A2/A₁R CAR T cells, overall cytokine produced remained higher than control CAR T cells for all three cytokines tested (**Supp Fig 8A**). We also demonstrated that the effects of enhanced cytokine could be reversed with A₁R antagonist (DPCPX), which showed no such effects on control or A_{2A}R KO CAR T cells (**Supp Fig 8B**). Supporting the reviewer's comment, we also show evidence of synergy between A_{2A}R deletion and A₁R induction, with these cells showing the highest level of cytokine secretion, and resistance to NECA suppression. In the instance of TNF and IL-2, NECA instead boosted the secretion of these cytokines in A_{2A}RKO+NR4A2/A₁R CAR T cells (**Reviewer Figure 2**).

To validate our observations suggesting differences between A_{2A}R KO and A₁R knock in, tumor stimulated NR4A2/A₁R and A_{2A}RKO CAR T cells treated with or without NECA were sequenced by RNA-seq to determine the transcriptional impacts of A₁R and A_{2A}R deletion in the context of adenosine receptor signaling. Principal component analyses of sequenced samples identified a clear separation mainly across Principal Component 1 (PC1, 64.1% variance explained) between NR4A2/hA₁R CAR-T cells compared to A_{2A}RKO or NR4A2KO controls (**Fig 5A**).

NECA treatment itself instead led to separation mainly across Principal Component 2 (PC2, 12.8% variance explained) in both NR4A2/hA₁R and NR4A2KO CAR-T cells, which was not observed in A_{2A}RKO CAR T cells, highlighting that NECA was mainly acting on the A_{2A}R in both NR4A2/hA₁R and NR4A2KO controls (**Fig 5A**).

Furthermore, there was minimal transcriptional differences between A_{2A}RKO and NR4A2KO control cells in the absence of NECA (18 DEGs up/down), whereas A₁R inducible expression

led to major transcriptional changes (1683 DEGs up/ 1809 DEGs down) versus NR4A2KO controls (**Fig 5B**). In the presence of NECA during tumor stimulation, there was very little overlap between transcriptional changes across NR4A2/hA₁R, A_{2A}ARKO and NR4A2KO CAR-T cells (**Fig 5B**), highlighting the unique transcriptional pathways associated with A₁R signaling. Finally, we demonstrate that the A₁R gene signature identified previously was only significantly enriched in NR4A2/hA₁R but not A_{2A}ARKO CAR-T cells, highlighting that A₁R expression and A_{2A}R deletion target unique transcriptional pathways to enhance CAR-T cell efficacy (**Fig 5C**).

In summary, these data indicate that A_{2A}R knockout and A₁R overexpression enhance CAR T cell function through distinct mechanisms. We have included data with dual A_{2A}R KO / A₁R knock in as a reviewer only figure (**Reviewer only Figure 2**) as we believe a comprehensive study of this engineering approach is beyond the scope of our current manuscript.

To clarify, we have demonstrated that:

1. A₁R expression drove effector differentiation, but not A_{2A}R deletion. This is linked to unique transcriptional changes.
2. NR4A2/A₁R engineered CAR T cells remain susceptible to A_{2A}R mediated suppression and thus combined A₁R knock-in and A_{2A}R KO leads to further enhancement of CAR T function.

An additional results section has been added to highlight this new data:

- See new results section between lines 571- 605

4. In Figure 3, please show the knock-in efficiency of multiple experiments.

Response:) In our hands, knock-in efficiency in human T cells ranges between 50-80%. These data are currently shown for representative *in vitro* and *in vivo* experiments in **Supplementary Figure 5B-D**. To address the reviewer's comment we have provided a **Reviewer Only Figure 3** demonstrating this knock-in efficiency was achieved across three different donors using the marker gene NGFR. Although we were unable to confirm knock-in efficiency with the A₁R construct, based upon the size of the A₁R (982bp) and NGFR (864bp) inserts a similar efficiency is anticipated. Furthermore, we have generated two independent bulk RNA-seq repeats demonstrating induction of A₁R at the transcriptional level after stimulation (**Figure 3C**).

Reviewer Figure 3: Knock-in efficiency of NGFR reporter in multiple independent donors. NGFR reporter was knocked-in at the NR4A2 locus of anti-LeY CAR T cells. CAR-T cells were chronically stimulated with OVCAR tumors for 72 hours and stained for NGFR expression. Data shown is shown as percentage positive of viable T cells representative of three independent healthy donors from independent experiments.

5. How long does A_{2A}R expression persist after antigen stimulation? Supp. Figure 5 only shows its upregulation 16 hours after stimulation. In addition, is it constitutively expressed in the tumor-infiltrating CAR-T cells in vivo because they are chronically exposed to the antigen?

Response: For clarification **Supp Figure 5** pertains to NR4A2 expression not A_{2A}R. In this study, we have knocked-in the A₁R into the NR4A2 loci and demonstrated in **Supplementary Figure 1C-D** that A_{2A}R is not modulated by A₁R expression. In the original manuscript, we have demonstrated in **Supp Figure 5**, specific expression of NR4A2 linked reporter gene by CAR T cells isolated from the tumor and not the spleen. A separate in-depth manuscript from our lab is under review at *Nature* highlights the pattern of transgene expression of a GFP reporter gene driven by the NR4A2 promoter. Please see excerpt from that paper included in **Reviewer Figure 4**. Critically NR4A2 is not active in healthy tissues, and is induced only in the tumor. We have also included analyses of NR4A2 reporter expression *in vitro* after chronic antigen exposure in **Reviewer Figure 5 and Supp Figure 5C**, demonstrating specific inducible expression only with CAR-T cell stimulation. To clarify this point we have inserted the following text at lines 493- 498 of the revised manuscript.

*“To highlight the potential for this approach to couple transgene expression to CAR activation, we first cocultured anti-Lewis Y CAR T cells engineered to express NGFR under the control of the NR4A2 promoter with Lewis Y⁺ tumor cells. In line with our previous work, NGFR expression was only observed following CAR activation and chronic stimulation (**Supp Fig 5B-C**) and assessment of A₁R mRNA in NR4A2/A₁R engineered CAR T cells confirmed that A₁R expression was only observed post CAR activation (**Fig 3C**).”*

Reviewer Figure 4. Transgene expression driven by the NR4A2 promoter *in-vivo*. Concatenated flow cytometry plots and quantification of GFP percentage in CAR T cells in tumor, spleen, liver, lung, brain, kidney and bone marrow of mice bearing orthotopic E0771-HER2 tumors 9 days post transfer.

Reviewer Figure 5. Inducible NGFR expression with chronic antigen stimulation after 72 hours. LeY-BBz CAR T cells were cocultured with LeY expressing OVCAR tumor cells or plate-bound α CD3+CD28 for 24 and 72 hours.

6. A1R upregulation under the control of the physiological NR4A2 promoter will be lower than that driven by a conventional retroviral promoter even after antigen stimulation, which may be a reason for the attenuated terminal differentiation and improved persistence of CAR-T cells. The authors should directly compare A1 receptor expression at the protein level.

Response: The reviewer has posed an interesting question. Whilst we have not tested constitutive expression of the A₁R using a less potent promoter, we have previously generated CAR T cells that express A₁R downstream of a synthetic NFAT promoter. However, the NFAT promoter system is more “leaky” and less specific than knock in to the NR4A2 locus, leading to low level transgene expression in peripheral sites (please refer to **Reviewer Figure 4**). Using this strategy we observed that NFAT-A₁R CAR T cells still elicited reduced persistence relative to control CAR T cells (**Reviewer Figure 6**). From this we infer that the kinetics and specificity of A₁R expression to the tumor site is more critical than the magnitude of expression. As outlined above in response to point 2 it is technically challenging to measure A₁R protein expression so we have not compared levels directly.

Reviewer Figure 6. NFAT promoter driven expression of A₁R in mouse CAR T cells partially reverses loss of short-term persistence in spleen and promotes tumor cell numbers in vivo. C57BL/6 Her2 transgenic mice were injected orthotopically with 2x10⁵ E0771-HER2 breast cancer cells in the fourth mammary fat pad and following total body irradiation of 4Gy were treated on days 7 and 8 post inoculation of tumors with 1x10⁷ Ctrl, A₁R, NFAT-GFP and NFAT-A₁R mouse anti-HER2 CAR T cells. Mice were treated with subcutaneous 50,000 IU of IL-2 each day on days 0-4 post treatment. A. Excised tumor weights (mg) at day 7 post therapy. Numbers of B. NGFR+ CD8+ and C. NGFR+ CD4+ CAR T cell counts in the spleen. Tumor infiltrating D. Numbers of NGFR+ CD8+ and E. NGFR+ CD4+ CAR T cell counts normalised to per mg of tumor weights. All data shown as means ± SEM of n=4-6 mice per group. ****p<0.0001, ***p<0.001, **p<0.01, *p<0.05, one-way ANOVA

7. In Figure 4a-c, the authors claim that tumor progression was better controlled by NR4A2/hA1R CAR-T cells. However, tumor growth was also suppressed in most of the mice

treated with control CAR-T cells and CAR-T cells with NR4A2 knockout (red and blue lines). In addition, there was no significant difference in the CD8⁺ T cell counts between control and NR4A2/hA1R CAR-T cells.

The authors should optimize a protocol to precisely determine the antitumor effects of NR4A2/hA1R CAR-T cells (e.g., reducing the CAR-T cell dose or infusing CAR-T cells at later time points). In addition, a longer follow-up period would be needed to evaluate the persistence of CAR-T cells.

Response: It is important to note that the anti-tumor effect of A₁R CAR T cells in the OVCAR tumor model as shown in **New Figure 4B** was significantly better than either control CAR T cells or NR4A2 KO CAR T cells. Nevertheless, we have examined the efficacy of A₁R CAR T cells in another CAR T cell model targeting HER2^{low} MDA-MB231 breast cancer to show the broad utility of our approach (please see comments to reviewer 1 point 4) (**New Figure 4A**). In this model, control CAR-T cells do not effectively control tumor growth, however A₁R CAR T cells show marked enhancement in tumor control.

8. The authors should confirm that IRF8 expression is increased by A1R expression and diminished upon CRISPR/Cas9 knockout at the protein level. What was the efficiency of the knockout as assessed at the genomic level?

Response: We have included qPCR data in the original manuscript confirming deletion of IRF8 in the **Supplementary Figure 9B**. We tested several IRF8 flow cytometry antibodies, and while we did see a decrease in IRF8 expression with CRISPR deletion, we found that the antibodies tested were not very sensitive. Therefore, we did not include this data in the manuscript. Please see **Reviewer Figure 7** below.

Reviewer Figure 7. Intra-nuclear flow cytometry staining of IRF8. Anti-LeY CAR T cells were cocultured with OVCAR tumor cells for 24 hours. NR4A2/hA₁R or NR4A2/hA₁R/IRF8KO CAR T cells were stained for intra-nuclear expression of IRF8.

9. The IRF8-knockout data are not convincing. While they did not affect cytokine secretion by CAR-T cells in Supp. Figure 8C, it was significantly suppressed in Figure 5I. The authors should provide a more convincing explanation for these different results. Also, CD62L⁺ CD69⁻ cells cannot be defined as memory T cells.

Response: We apologise for the misunderstanding. The original **Figure 5I (new Figure 5L)** refers to A₁R expressing cells and **Supp Figure 8C (New Supp Figure 9C)** refers to control cells. We demonstrate that IRF8 KO reverses the A₁R phenotype (**Figure 5L**), but has no significant effect on cytokine production in control CAR T cells (**New Supp Fig 9C**). This is consistent with the fact that IRF8 was significantly upregulated in A₁R CAR T cells and identified as a key transcription factor for the A₁R phenotype (**Fig5 I-K**), whilst it does not appear to have a role in control CAR T cells. We have further highlighted this point in our revised version of the paper, see lines 631-634.

“CRISPR-mediated deletion of IRF8 (Supp Fig 9B) in control human anti-Lewis Y CAR T cells did not significantly modulate cytokine production or expression of memory and exhaustion markers TCF-1, TIM-3 or PD-1 in unstimulated or tumor stimulated CAR T cells (Supp Fig 9C-E).”

10. Most critically, what is the overall benefit of the strategy of expressing A1R under the NR4A2 promoter compared to the simple knockout of the A2A receptor presented in the previous paper? Since CRISPR/Cas9-mediated gene knock-in is a more complicated procedure, it will not be practically useful unless it has some functional advantage over the knockout of A2A receptor. It will be necessary to directly compare the *in vivo* efficacy, persistence and representative phenotypes of CAR-T cells between control, A1 receptor overexpression, NR4A2-A1R expression, and A2A receptor KO CAR-T cells.

Response: Please see responses to question 3 (Reviewer 2) above. To compare A₁R knock in and A_{2A}R knock-out strategies, we undertook new functional experiments and bulk RNAseq analysis of A₁R expressing vs A_{2A}RKO CAR T cells. Our new data indicate that A₁R has unique effects on CAR T cells independent of A_{2A}R and the two engineering strategies infer distinct transcriptional programs. Interestingly our preliminary analyses revealed synergy in combination A_{2A}RKO and inducible A₁R CAR T cells. We believe that an *in vivo* assessment of this combination approach is beyond the scope of the current paper but we agree with the reviewer that this is an interesting avenue for further exploration.

Reviewer #3 (Remarks to the Author): with expertise in CAR-T cells

I really liked the premise for your paper based on your earlier work. And a really cool advance to limit A1R expression to T cells traversing the TME.

In this article, Sek and coworkers engineer CAR T cells to express A1R, a receptor that opposes the immune-suppressive effects of A2R by dampening levels of the second messenger cAMP. In a comprehensive and extensive analyses (across murine and T cells), constitutive A1R expression drives maximal effector function, albeit at the expense of persistence. This tradeoff can be overcome by limiting A1R expression until CARTs traverse the TME. This innovative advance in site-directed expression led to enhanced anti-tumor function. Compelling mechanistic insights from this study reveal that IRF8 is necessary to drive the enhanced anti-tumor benefit from A1R. This manuscript is well written, articulate, and highly insightful for leveraging T cell differentiation, effector function and persistence following adoptive transfer. It will also appeal to the immunometabolism community at large, as it provides an important tool to skew differentiation/keep it at bay to retain memory cell precursors.

Addressing the following comments will make it a stronger manuscript:

Response: We would like to thank the reviewer for their favourable review of our article.

Major points

1) What is endogenous A₁R expression in naïve, Tcm, and effector T cells? Given its role in effector function, it would be interesting to see if it is induced in effector differentiated cells.

Response: A₁R is not reported to be expressed endogenously in T cells (PMID 11992407) and we also did not observe A₁R mRNA expression in qRT-PCR or RNA-Seq analysis of control CAR T cells and so we were unable to compare expression in different T cell subsets.. We have included a statement to this effect in the revised version of the paper. See lines 666-667.

“Whilst A₁R is not expressed on lymphocytes physiologically, previous studies have suggested a role for A₃R in promoting T cell effector functionality.”

2) How do intracellular cAMP levels change during T cell differentiation (naïve, Tcm, Tem)?

Response: cAMP signalling is a function of extracellular metabolites (predominantly PGE₂ and adenosine) rather than at a T cell differentiation stage. A related relevant question is the expression of the key receptors e.g. A_{2A}R that drive cAMP signaling. Expression of A_{2A}R is increased on activated T cells e.g. T_H1 (PMID 10051547) and elevated on all subsets within the tumor microenvironment (Todd *et al.* Nature Communications 2023). We have referred to this study and other prior literature in the revised paper. See lines 69-71.

How does the anti-tumor function of A₁R-expressing CAR T cells compare to CAR T cells that have A_{2R} deleted (in the same experiment)? It would be good to compare.

Response: Please refer to our responses to questions 3 and 10 (Reviewer 2) above.

How does IRF8 expression change during your chronic stimulation assay? Does its expression coincide with TNFalpha abundance which declines dramatically after the third successive stimulation.

Response: The reviewer has raised an interesting question. We observed that IRF8 transcriptionally increases in the first stimulation and slowly decreases over time as highlighted in **Figure 5K**. Its pattern of expression does coincide with TNF but it is difficult to ascribe a causal relationship.

Additionally, how do intracellular cAMP levels change as T cells undergoing serial stimulation? Is there a dramatic increase after stimulation #3?

Response: Performing cAMP assays on stimulated T cells is technically challenging as the assays needs to be optimised for each cell type used. Please refer to the optimisation data presented in response to reviewer 2 point 1. We therefore do not propose to test this experimentally given that we do not believe that modification of cAMP signaling is a major mechanism for the A₁R mediated effect.

Does IRF8 alter metabolic fitness or is it just linked to differentiation state exclusively? Is it established if adenosine is high in the E0771-Her2 tumor model (Fig 2A)?

Response: While this is a very interesting question, we have not interrogated the role of IRF8 in this context because metabolic pathways were not observed to be significantly modulated by A₁R expression. For example the glycolysis pathway was most significantly associated with the Turquoise gene module (**Figure 5E**), which was not significantly different between control and A₁R expressing CAR T cells (**Supplementary Figure 9A**). However we have added lines 731-732 to the discussion, highlighting a potential role for IRF factors as metabolic stress sensors.

“Additionally, IRF factors have been implicated in metabolic processes as stress sensors, like other factors such as Foxo1, which could be manipulated to further improve CAR-T cell persistence”

Regarding E0771-Her2, it is technically challenging to assess adenosine levels *ex vivo* due to its short half life and we do not have the capacity to do this. Therefore, whilst we cannot answer this question directly, our previous studies have indicated that T cells infiltrating E0771 tumors express high levels of the A_{2A}R and that CAR T cells lacking the A_{2A}R exhibit enhanced anti-tumor activity against E0771-Her2 tumors *in vivo* (Giuffrida, Sek *et al.* 2021, Beavis *et al.* 2017).

Minor Points

It is a pity that transient inhibition of A₁R during expansion (prior to adoptive transfer), didn't enhance anti-tumor function and T cell persistence. How does this reconcile with the corresponding benefit from the induction of A₁R within the tumor?

Response: This is likely explained by the transient nature of A₁R inhibition provided by a pharmacological agent, which is lost upon adoptive transfer. However, NR4A2 regulation means that A₁R is not induced until T cell antigen stimulation by the tumor, therefore overcoming this limitation. We have included a statement in the revised paper to address this point. See lines 738-740 and lines 745-749.

“To overcome this, we utilized a novel CRISPR/Cas9 knock-in strategy to drive A₁R expression specifically at the tumor site, leading to enhanced therapeutic efficacy and comparable engraftment, persistence, and expansion.”

“This highlights the importance of preserving memory but enhancing effector differentiation, and is to our knowledge, the first demonstration of an approach to engineer CAR T cell differentiation states specifically at the tumor site. Importantly this approach can be applied to any receptor or transcription factor that promotes short-term CAR T cell functionality at the expense of long-term persistence.”

Reviewer #4 (Remarks to the Author): with expertise in cancer immunology, adenosine signalling

In this extensive and impressively illustrated study, the authors report that engineering of CAR T cells with A₁R enhances their efficacy against solid tumors *in vitro* and *in vivo*. This is a highly relevant study that could impact therapy of solid tumors with CAR T cells. The latter has not been as satisfactory as expected: only a small percentage of patients treated with CAR T cells achieve CR. This lack of efficacy is attributable to various barriers, but the authors focus on ADO-mediated immune suppression as the major check point for tumor-directed CAR T cells.

Here, they describe “arming” of CAR T cells against immunosuppressive TME by overexpression of A1R, which attenuates suppressive effects of A2AR, a major ADOR in T cells, by inhibition of adenylate cyclase and cAMP accumulation. The authors show that overexpression of A1R and not of A3R suppressed cAMP production and enhanced IFN γ , TNF and granzyme activities in both murine and human CAR T cells. However, constitutive A1R expression resulted in a loss of CAR T cell persistence and reduction in the T stem cell memory (TSCM) subset. So, the authors opted for transient A1R expression by restricting A1R signaling to the tumor site, where ADO is produced. Restriction of A1R to the tumor site was accomplished by using a novel CRISPR knock-in approach under control of the NR4A2 promoter. This CAR T cell modification drove their differentiation and promoted anti-tumor functionality without having any impact on T cell persistence. Thus, the ability to transiently overexpress A1R at the tumor site led to improved therapeutic efficacy by activation of T cells within the tumor but not the spleen. Overexpression of A1R restricted ADO produced by the tumor and reduced ADO mediated suppression of CAR T cells. While the experimental design is rational and experiments were rigorously controlled, there are some concerns. These largely consists of a potential transfer of this methodology to the clinic.

Response: We would like to thank the reviewer for their positive review of our article.

First, knock-in of A1R gene, while feasible, might alter the signaling balance of ADORs in T cells as well as non-immune cells in an unexpected way. Are there any off target effects that the authors observed?

Response: It is not entirely clear what is meant by off target effects in this context. We have further included safety data in the form of mouse weights in **Supp Figure 7B** for a second xenograft mouse tumor and CAR-T model, which infers that there is no off-target toxicity of our approach. With regards to potential off target effects e.g. to non immune cells as a result of our CRISPR engineering strategy, it is important to highlight that the *ex vivo* engineering approach used in our study ensures that edits are specifically made to the CAR T cell population. One concern with regards to CRISPR editing is the potential for off target deletions. To generate an sgRNA with low potential for off target edits we used the COSMID web based tool (PMID 25462530), which indicated no potential off target editing sites for this sgRNA. We agree that for clinical application of this approach further vigorous testing of this sgRNA would be required such as whole genome sequencing as used for sgRNA that have been used to generate products for clinical evaluation. However, this is beyond the scope and resources of this proof of concept preclinical study and even if off target edits were associated with this particularly sgRNA, another could be designed to achieve the same biological effect.

Another concern transient effects of relates to A₁R site-specific overexpression; how will timing of adoptive transfers be orchestrated to achieve optimal therapeutic effects?

Response: In this system CAR T cells upregulate A₁R when they encounter tumor antigen and so the timing of adoptive transfer is inconsequential to this mechanism. However the reviewer eludes to an important question with regards to the optimal magnitude and duration of A₁R expression. This could be optimised through the use of alternative promoters with higher/ lower transgene expression or different kinetics of expression. These investigations are of interest but beyond the scope of the current work.

Also, the authors are overly optimistic about therapeutic efficacy of this therapy targeting but a single suppressive pathway. Although the ADO may be the major checkpoint in some HNCs, it is not the only one. Numerous other immunosuppressive pathways exist, and may differ in functional intensity among patients. Is blockade of one pathway be sufficient to save CAR T cells from tumor-induced suppression??.

Response: We agree with the reviewer that enabling effective treatment of CAR T cells with solid tumors is a significant challenge but this approach demonstrates that it is possible to tune CAR T cell effector differentiation to occur at the “right” time. Importantly, A₁R promotes effector function independently of adenosine mediated suppression, suggesting this approach could be applicable even in situations where adenosine mediated suppression is not prevalent. We agree with the reviewer that there may be other synergistic combinations with A₁R inducible expression that could further enhance CAR-T cell therapy. We are currently exploring other combinations in future studies, including further synergistic targeting of the ADO pathway by A_{2A}R deletion, combination with immune checkpoint inhibitors, and other approaches to promote CAR-T cell memory, persistence and engraftment. To address this point we have included further discussion of this point on line 739-749.

“Several studies have demonstrated the feasibility of favorably controlling CAR T cell differentiation through controlled programming to promote anti-tumor function, most of which have focused on promoting stemness features through inosine [87] or cytokine preconditioning [59, 92, 93] or overexpression of memory associated transcription factors such as FOXO1 [91, 94]. Others have targeted factors that drive or prevent exhaustion such as c-Jun [70], c-Myb [95] or BATF [58]. This highlights the importance of preserving memory but enhancing effector differentiation, and is to our knowledge, the first demonstration of an approach to engineer CAR T cell differentiation states specifically at the tumor site. Importantly this approach can be applied to any receptor or transcription factor that promotes short-term CAR T cell functionality at the expense of long-term persistence.”

The manuscript is clearly written, and the data presentation is excellent.

Response: We thank the reviewer for their positive feedback.

Point by Point Response:

Reviewer #1 (Remarks to the Author): with expertise in CAR-T cells

Sek et al have submitted a manuscript where they investigate the role and potential of A1AR signaling in CAR T cell function in relationship to suppressive A2AR receptors. They show that while A1AR signaling may enhance cytokine production capabilities in vitro, on the long run this positive effect is countered by A2AR signaling which seem to drive T cell differentiation away from SCM T cells. Thus simultaneous knock of of A2AR and A1AR expression circumvents this effects and results in bettered outcomes in vivo, due to IRF8 signaling. The manuscript is well written and mostly sound. The actual novelty comes from the use of A1AR, preferably in the absence of A2AR, the rest of the techniques being known.

Response: We thank the reviewer for their positive critique of our paper. We believe that the reviewer may also be referring to our data where A₁R was knocked in to the NR4A2 locus rather than a ‘simultaneous knock-out’ of A_{2A}R. This was done in order to achieve tumor-site specific expression of A₁R.

I have the following concerns:

1) Overall I do not agree with the notion that A1AR expression in the absence of A2AR KO would be worth then CAR in vivo. This is not backed up by data and should be corrected to non-different.

Response: Our original manuscript did not report data with A_{2A}R knock-out but rather NR4A2 knockout. We apologise for this confusion, and we have sought to clarify this further in the text and in **Supp Figure 5A**.

See additional text: *“Furthermore, it is important to note that this approach simultaneously inserts the A₁R gene while knocking out NR4A2 (Supp Fig 5A).”* in lines 491-493.

2) It is surprising that while A2AR has a known suppressive effect on CAR T cells. Knock out thereof does not seem to lead to expected outcome. This needs further elaboratio

Response: Please see response to Reviewer 1 point 1) above.

3) The amount of TSCM after CAR T cell generation seem quite high with 20+%, how is this protocol compared to regular expansion protocols typically at best in the single digit range?

Response: Our expansion protocol is based upon previously published work from our labs and others, and our results are consistent with these studies (Fraieta *et al.* 2018 *Nature Medicine*, Chan *et al.* 2024, *Nature*). Our TSCM gating strategy based off these two publications is shown in **Supp Figure 3B** (Mouse) and **Supp Figure 3C-D** (Human).

4) A major concern is the focus on a single model to demonstrate utility: OVCAR, which is very CAR permissive. Especially in the context of comment 1). More evidence in different models and ideally with different CAR is needed to allow generalization of the conclusions. Otherwise this is a one model thing.

Response: We agree with the reviewer that data in a second model would strengthen our conclusions. In our revised paper, we evaluate the effect of inducible A₁R expression in an additional *in vivo* tumor model as well as *in vitro* analysis of 3 alternative CAR T cell antigen systems using both retroviral and lentiviral transduction protocols. This additional data has now been included as additional panels in **Supp Fig 6D-F** and **Figure 4A**.

- See lines 513-522 in Results section:

*“To evaluate the broad applicability of NR4A2/hA₁R to enhance T cell adoptive therapy, we also applied this engineering approach to CARs targeting the HER2 and ROR1 antigens and in the context of CARs with either a CD28 or 4-1BB signaling domain. Consistent with our results obtained with anti-Lewis Y CAR T cells, NR4A2/A₁R engineered anti-HER2 CAR T cells exhibited significantly enhanced cytokine production upon coculture with HER2⁺ MCF-7 breast tumors (Fig 3I, Supp Fig 6C). Regardless of costimulatory domains expressed, HER2-CARs with CD28 or 41BB domains secreted enhanced cytokine production against MDA-MB231 breast tumors (Supp Fig 6D-E). Similarly, NR4A2/hA₁R engineering enhanced the cytokine production of anti-ROR1 CAR T cells when cocultured with MDA-MB231 tumors *in vitro* (Supp Fig 6F).”*

- Enhanced *in vivo* efficacy and survival in a HER2 breast cancer model by A₁R edited anti-HER2 CAR T cells (**Figure 4A**). This data is referred to in lines 545-547 of the text:

*“To determine the *in vivo* efficacy of NR4A2/hA₁R engineered CAR T cells, 5x10⁶ anti-HER2 CAR T cells or 15x10⁶ anti-Lewis Y CAR T cells were adoptively transferred to treat MDA-MB231 breast (Fig 4A) or OVCAR-3 ovarian (Fig 4B) tumor bearing mice respectively.”*

5) Increased cytokine secretion is not universally good and can be actually quite detrimental in terms of safety. What is the contribution of cytokines to the observed effects? The SCM relationship would typically be more driven towards T cell fitness.

Response: The reviewer has highlighted a recent important finding in the CAR-T field which has found that anti-tumor cytokines are potentially dispensable in liquid tumors, however they are critical for an effective response in the context of solid tumors as demonstrated by Larson *et al.* 2021, *Nature*. (PMID 35418687). We agree with the reviewer that cytokine secretion by CAR T cells can be detrimental in certain contexts, but this is more commonly associated with hematological cancers (e.g. PMID 37296093). Furthermore, we have also previously shown this to be the case following IFN γ blockade in our solid tumor models following adoptive transfer of CAR T cells as used in the current study (Beavis *et al.* *JCI* 2017; Chan *et al.* *Nature*, 2024). Assessment of the requirement for enhanced cytokine production in the anti-tumor efficacy for NR4A2/A₁R engineered CAR T cells was beyond the scope of our study but we have provided additional discussion on this point on line 91-94 of the revised manuscript:

“In this work, we show that expression of A₁R but not A₃R suppresses cAMP production in CAR T cells and enhances both murine and human CAR T cell function by promoting their production of IFN γ and TNF, cytokines that have been demonstrated to be critical for anti-tumor efficacy of CAR T cells in solid tumors [19, 29, 30].”

6) For each and every figure and subfigure the amount of independent repeats and replicates as well as donors must be clearly stated.

Response: This information has now been included in the revised manuscript

7) How was cell line identity ascertained.

Response: We utilize short tandem repeat (STR) analyses to confirm the identity of our cell lines and used within 10 passages of a master stock to ensure their accuracy.

Reviewer #2 (Remarks to the Author): with expertise in CAR-T cells

In this manuscript, Sek et al. ectopically expressed A1R to overcome adenosine-A2AR-mediated T cell exhaustion. The authors first tested A1R overexpression and found that this strategy did not improve overall antitumor immunity due to impaired T-cell persistence. They then investigated the effect of activation-dependent A1R expression by knocking in the gene under the NR4A2 promoter. The authors showed that this strategy can overcome the problem of constitutive A1R expression. I have the following specific comments.

1. In Figure 1B, the authors analyzed cAMP levels only under artificial experimental conditions: exogenous addition of adenosine to forskolin-stimulated T cells). Please show the data that A1R overexpression can reduce intracellular cAMP levels in a more physiological setting (e.g., antigen-stimulated CAR-T cells without adenosine treatment).

Response: We thank the reviewer for this question - it raises some interesting points around 1. the activity of forskolin at adenylate cyclase and 2. some sensitivity aspects of the assay.

1. Forskolin is a small molecule activator of adenylate cyclase that acts in synergy with GalphaS proteins via a structurally distinct site (PMID 38351373). Forskolin acts to potentiate the activity of GalphaS (in this case working via adenosine-mediated activation of the A_{2A}R) and can increase the rate of conversion of ATP to cAMP. The use of a sub-maximal concentration of forskolin will 'prime' the adenylate cyclase for activation by GalphaS post-A_{2A}R activation. Together, in lieu of high A₁R constitutive activity, the addition of exogenous ADO and concomitant A₁R and A_{2A}R activation and forskolin-priming allows us to demonstrate the modulatory effect of the A₁R on T cell cAMP.

2. The design of the cAMP kit is a competition assay between cAMP in cell lysates and a cAMP probe, which means the response has an upper and lower limit based on the amount of antibody and probe used in the assay (recommended to not alter reagent ratios) and that the cAMP in lysates needs to be in the dynamic range of the system (0.5-15nM), which is why we use a forskolin-priming approach and ADO to raise the level of cAMP into that range. Low local levels of cAMP still have big impacts on biological outcomes, but may well be below detectable limits with the technologies available. Hence, we feel our assay design gives a correct appraisal of the ADO-mediated cAMP dynamics and the overall effect of A₁R overexpression and activation.

We have further included **Reviewer Figure 1** below for the optimisation of the cAMP assay. We show that the priming concentration of forskolin (1µM) was selected so as to not drive cAMP in primary T cells on its own, and that this priming was essential for achieving the dynamic range in this assay when NECA or ADO was added.

Thus, we are confident that our data indicates that A₁R expression can inhibit A_{2A}R mediated cAMP signaling. We also note that our data indicates that A₁R knock-in induces a distinct transcriptional program relative to A_{2A}R KO (please refer to response to Point 3 Reviewer 2), suggesting that modulation of cAMP signaling is only one component by which A₁R expression enhances CAR T cell effector function.

Reviewer Figure 1. Optimization of Lance Ultra cAMP assay for detecting intracellular cAMP in CAR T cells. Control CAR T Cells at days 7-9 post transduction were incubated with Forskolin (FSK) for 30 minutes before internal cell lysates were used to detect cAMP. A. TR-FRET lance output based on the standard curve of known concentrations of cAMP B. TR-FRET lance output of 1,000-20,000 CAR T cell numbers incubated for 30 minutes with increasing concentrations of FSK. C. cAMP readout of 5000 CAR T cells after incubation with FSK for 30 minutes. D. Control CAR T cells were incubated with/without FSK (1 μ M) for 30 minutes, before NECA was added at varying concentrations for another 30 minutes. Triplicate means \pm SD from n=1 experiment ****p<0.0001, *p<0.05, One-way ANOVA

2. Please show cell-surface expression of the A₁ receptor in control and overexpressing T cells analyzed by flow cytometry. Several antibodies are commercially available.

Response: In our experience, antibodies against adenosine receptors lack specificity and sensitivity (which is why we previously generated an A_{2A}R reporter mouse; Todd *et al.* 2023 Nature Communications). Although we were unable to evaluate potential endogenous A₁R

protein expression, our mRNA data indicate that control T cells express almost undetectable levels of A₁R mRNA, a result that has previously been reported by other groups (PMID 11992407).

However, to address this question, we have labelled the hA₁R with a MYC tag while the anti-LeY CAR is labelled with a FLAG tag. Utilizing both tags, we demonstrate constitutive co-expression of A₁R and CAR proteins on the cell surface of lentivirally transduced primary human T cells. Furthermore, CD69 upregulation by constitutive A₁R expression was consistent with our untagged A₁R construct indicating a functional A₁R phenotype. This data has been included as a panel in our modified paper **Supp Fig 2C-D** and referred to in the following text on lines 374-378. The full engineered myc-tag hA₁R protein sequence is provided in **Supp Table 1**.

“Due to the lack of reliable flow cytometry antibodies for the hA₁R, a short Myc-tag sequence was engineered onto the extracellular region of the wild-type hA₁R to confirm protein expression on the cell surface (Supp Fig 2C, Supp Table 1). Co-expression of Flag-tag CAR and Myc-tag hA₁R was determined by flow cytometry, confirming successful cell surface expression of A₁R on anti-Lewis Y CAR T cells (Supp Fig 2C-D)”.

3. In the previous paper by the same group (Giuffrida et al. Nat Commun 2021), the authors showed that genetic ablation of ADORA2A enhances both effector functions and persistence of CAR-T cells in mouse tumor models. The data presented in Figure 2 of this manuscript appear to be inconsistent with these findings. How do the authors reconcile this discrepancy? If it is due to a different experimental protocol, the authors should confirm whether A_{2A}R-knockout CAR-T cells show similar profiles in the present protocol (accelerated differentiation and poor persistence). Another possibility is that A₁R overexpression may affect T cell function by other mechanisms in addition to suppression of A_{2A}R-mediated signaling.

Response: We thank the reviewer for their insightful comments and we believe that the reviewer is correct that A₁R is affecting T cell function through mechanisms other than suppression of A_{2A}R mediated signaling. Therefore, the data shown in Figure 2 should not be considered a discrepancy in relation to previous experiments with A_{2A}R KO CAR T cells, which exhibit a distinct phenotype based on the different mechanisms of the two engineering strategies.

Indeed, the reviewer has raised important questions on the overlap in signaling pathways between the A_{2A}R and A₁R, and the potential for synergy between A_{2A}R deletion and A₁R tumor-site specific expression.

To address this we have therefore performed an *in vitro* assessment of function and transcriptional activity of A₁R knock-in vs A_{2A}R knockout cells in the presence of the adenosine analogue NECA or A₁R inhibitor DPCPX to demonstrate the differences between the two approaches. Furthermore, we have combined A_{2A}RKO and NR4A2/A₁R, demonstrating potential synergy *in vitro* (**Reviewer Figure 2**).

Reviewer Figure 2. A₁R expression and A_{2A}R deletion in CAR T cells differentially regulate cytokine expression. Cytokine production in engineered CAR T cells that were serially stimulated with OVCAR tumors for 72 hours in the presence or absence of pan-adenosine receptor agonist NECA (10 μ M) normalised against NR4A2 CAR-T cells without NECA control. **** p <0.0001, *** p <0.001, ** p <0.01, * p <0.05. (a, b) two-way ANOVA. Data represented as mean \pm SD from 2 independent donors.

We have included the new data in **Supp Figure 8**, showing that NR4A2/A₁R expression drives significantly enhanced cytokine production of IFN γ and TNF upon CAR T cell activation, which was not observed with A_{2A}R deletion (**Supp Fig 8A**). Furthermore, we demonstrated that, consistent with our previous studies (Giuffrida, Sek *et al.* 2021, Beavis *et al.* 2017), a pan-adenosine receptor agonist, NECA, could suppress IFN γ and TNF in control CAR T cells, but that this effect was lost in A_{2A}R KO CAR T cells showing that suppression of cytokines is A_{2A}R mediated (**Supp Fig 8A**). Interestingly, while NECA suppressed IFN γ in NR4A2/A₁R CAR T cells, overall cytokine produced remained higher than control CAR T cells for all three cytokines tested (**Supp Fig 8A**). We also demonstrated that the effects of enhanced cytokine could be reversed with A₁R antagonist (DPCPX), which showed no such effects on control or A_{2A}R KO CAR T cells (**Supp Fig 8B**). Supporting the reviewer's comment, we also show evidence of synergy between A_{2A}R deletion and A₁R induction, with these cells showing the highest level of cytokine secretion, and resistance to NECA suppression. In the instance of TNF and IL-2, NECA instead boosted the secretion of these cytokines in A_{2A}RKO+NR4A2/A₁R CAR T cells (**Reviewer Figure 2**).

To validate our observations suggesting differences between A_{2A}R KO and A₁R knock in, tumor stimulated NR4A2/A₁R and A_{2A}RKO CAR T cells treated with or without NECA were sequenced by RNA-seq to determine the transcriptional impacts of A₁R and A_{2A}R deletion in the context of adenosine receptor signaling. Principal component analyses of sequenced samples identified a clear separation mainly across Principal Component 1 (PC1, 64.1% variance explained) between NR4A2/hA₁R CAR-T cells compared to A_{2A}RKO or NR4A2KO controls (**Fig 5A**).

NECA treatment itself instead led to separation mainly across Principal Component 2 (PC2, 12.8% variance explained) in both NR4A2/hA₁R and NR4A2KO CAR-T cells, which was not observed in A_{2A}RKO CAR T cells, highlighting that NECA was mainly acting on the A_{2A}R in both NR4A2/hA₁R and NR4A2KO controls (**Fig 5A**).

Furthermore, there was minimal transcriptional differences between A_{2A}RKO and NR4A2KO control cells in the absence of NECA (18 DEGs up/down), whereas A₁R inducible expression

led to major transcriptional changes (1683 DEGs up/ 1809 DEGs down) versus NR4A2KO controls (**Fig 5B**). In the presence of NECA during tumor stimulation, there was very little overlap between transcriptional changes across NR4A2/hA₁R, A_{2A}ARKO and NR4A2KO CAR-T cells (**Fig 5B**), highlighting the unique transcriptional pathways associated with A₁R signaling. Finally, we demonstrate that the A₁R gene signature identified previously was only significantly enriched in NR4A2/hA₁R but not A_{2A}ARKO CAR-T cells, highlighting that A₁R expression and A_{2A}R deletion target unique transcriptional pathways to enhance CAR-T cell efficacy (**Fig 5C**).

In summary, these data indicate that A_{2A}R knockout and A₁R overexpression enhance CAR T cell function through distinct mechanisms. We have included data with dual A_{2A}R KO / A₁R knock in as a reviewer only figure (**Reviewer only Figure 2**) as we believe a comprehensive study of this engineering approach is beyond the scope of our current manuscript.

To clarify, we have demonstrated that:

1. A₁R expression drove effector differentiation, but not A_{2A}R deletion. This is linked to unique transcriptional changes.
2. NR4A2/A₁R engineered CAR T cells remain susceptible to A_{2A}R mediated suppression and thus combined A₁R knock-in and A_{2A}R KO leads to further enhancement of CAR T function.

An additional results section has been added to highlight this new data:

- See new results section between lines 571- 605

4. In Figure 3, please show the knock-in efficiency of multiple experiments.

Response:) In our hands, knock-in efficiency in human T cells ranges between 50-80%. These data are currently shown for representative *in vitro* and *in vivo* experiments in **Supplementary Figure 5B-D**. To address the reviewer's comment we have provided a **Reviewer Only Figure 3** demonstrating this knock-in efficiency was achieved across three different donors using the marker gene NGFR. Although we were unable to confirm knock-in efficiency with the A₁R construct, based upon the size of the A₁R (982bp) and NGFR (864bp) inserts a similar efficiency is anticipated. Furthermore, we have generated two independent bulk RNA-seq repeats demonstrating induction of A₁R at the transcriptional level after stimulation (**Figure 3C**).

Reviewer Figure 3: Knock-in efficiency of NGFR reporter in multiple independent donors. NGFR reporter was knocked-in at the NR4A2 locus of anti-LeY CAR T cells. CAR-T cells were chronically stimulated with OVCAR tumors for 72 hours and stained for NGFR expression. Data shown is shown as percentage positive of viable T cells representative of three independent healthy donors from independent experiments.

5. How long does A_{2A}R expression persist after antigen stimulation? Supp. Figure 5 only shows its upregulation 16 hours after stimulation. In addition, is it constitutively expressed in the tumor-infiltrating CAR-T cells in vivo because they are chronically exposed to the antigen?

Response: For clarification **Supp Figure 5** pertains to NR4A2 expression not A_{2A}R. In this study, we have knocked-in the A₁R into the NR4A2 loci and demonstrated in **Supplementary Figure 1C-D** that A_{2A}R is not modulated by A₁R expression. In the original manuscript, we have demonstrated in **Supp Figure 5**, specific expression of NR4A2 linked reporter gene by CAR T cells isolated from the tumor and not the spleen. A separate in-depth manuscript from our lab is under review at *Nature* highlights the pattern of transgene expression of a GFP reporter gene driven by the NR4A2 promoter. Please see excerpt from that paper included in **Reviewer Figure 4**. Critically NR4A2 is not active in healthy tissues, and is induced only in the tumor. We have also included analyses of NR4A2 reporter expression *in vitro* after chronic antigen exposure in **Reviewer Figure 5 and Supp Figure 5C**, demonstrating specific inducible expression only with CAR-T cell stimulation. To clarify this point we have inserted the following text at lines 493- 498 of the revised manuscript.

*“To highlight the potential for this approach to couple transgene expression to CAR activation, we first cocultured anti-Lewis Y CAR T cells engineered to express NGFR under the control of the NR4A2 promoter with Lewis Y⁺ tumor cells. In line with our previous work, NGFR expression was only observed following CAR activation and chronic stimulation (**Supp Fig 5B-C**) and assessment of A₁R mRNA in NR4A2/A₁R engineered CAR T cells confirmed that A₁R expression was only observed post CAR activation (**Fig 3C**).”*

Reviewer Figure 4. Transgene expression driven by the NR4A2 promoter *in-vivo*. Concatenated flow cytometry plots and quantification of GFP percentage in CAR T cells in tumor, spleen, liver, lung, brain, kidney and bone marrow of mice bearing orthotopic E0771-HER2 tumors 9 days post transfer.

Reviewer Figure 5. Inducible NGFR expression with chronic antigen stimulation after 72 hours. LeY-BBz CAR T cells were cocultured with LeY expressing OVCAR tumor cells or plate-bound α CD3+CD28 for 24 and 72 hours.

6. A1R upregulation under the control of the physiological NR4A2 promoter will be lower than that driven by a conventional retroviral promoter even after antigen stimulation, which may be a reason for the attenuated terminal differentiation and improved persistence of CAR-T cells. The authors should directly compare A1 receptor expression at the protein level.

Response: The reviewer has posed an interesting question. Whilst we have not tested constitutive expression of the A₁R using a less potent promoter, we have previously generated CAR T cells that express A₁R downstream of a synthetic NFAT promoter. However, the NFAT promoter system is more “leaky” and less specific than knock in to the NR4A2 locus, leading to low level transgene expression in peripheral sites (please refer to **Reviewer Figure 4**). Using this strategy we observed that NFAT-A₁R CAR T cells still elicited reduced persistence relative to control CAR T cells (**Reviewer Figure 6**). From this we infer that the kinetics and specificity of A₁R expression to the tumor site is more critical than the magnitude of expression. As outlined above in response to point 2 it is technically challenging to measure A₁R protein expression so we have not compared levels directly.

Reviewer Figure 6. NFAT promoter driven expression of A₁R in mouse CAR T cells partially reverses loss of short-term persistence in spleen and promotes tumor cell numbers in vivo. C57BL/6 Her2 transgenic mice were injected orthotopically with 2x10⁵ E0771-HER2 breast cancer cells in the fourth mammary fat pad and following total body irradiation of 4Gy were treated on days 7 and 8 post inoculation of tumors with 1x10⁷ Ctrl, A₁R, NFAT-GFP and NFAT-A₁R mouse anti-HER2 CAR T cells. Mice were treated with subcutaneous 50,000 IU of IL-2 each day on days 0-4 post treatment. A. Excised tumor weights (mg) at day 7 post therapy. Numbers of B. NGFR+ CD8+ and C. NGFR+ CD4+ CAR T cell counts in the spleen. Tumor infiltrating D. Numbers of NGFR+ CD8+ and E. NGFR+ CD4+ CAR T cell counts normalised to per mg of tumor weights. All data shown as means ± SEM of n=4-6 mice per group. ****p<0.0001, ***p<0.001, **p<0.01, *p<0.05, one-way ANOVA

7. In Figure 4a-c, the authors claim that tumor progression was better controlled by NR4A2/hA1R CAR-T cells. However, tumor growth was also suppressed in most of the mice

treated with control CAR-T cells and CAR-T cells with NR4A2 knockout (red and blue lines). In addition, there was no significant difference in the CD8⁺ T cell counts between control and NR4A2/hA1R CAR-T cells.

The authors should optimize a protocol to precisely determine the antitumor effects of NR4A2/hA1R CAR-T cells (e.g., reducing the CAR-T cell dose or infusing CAR-T cells at later time points). In addition, a longer follow-up period would be needed to evaluate the persistence of CAR-T cells.

Response: It is important to note that the anti-tumor effect of A₁R CAR T cells in the OVCAR tumor model as shown in **New Figure 4B** was significantly better than either control CAR T cells or NR4A2 KO CAR T cells. Nevertheless, we have examined the efficacy of A₁R CAR T cells in another CAR T cell model targeting HER2^{low} MDA-MB231 breast cancer to show the broad utility of our approach (please see comments to reviewer 1 point 4) (**New Figure 4A**). In this model, control CAR-T cells do not effectively control tumor growth, however A₁R CAR T cells show marked enhancement in tumor control.

8. The authors should confirm that IRF8 expression is increased by A1R expression and diminished upon CRISPR/Cas9 knockout at the protein level. What was the efficiency of the knockout as assessed at the genomic level?

Response: We have included qPCR data in the original manuscript confirming deletion of IRF8 in the **Supplementary Figure 9B**. We tested several IRF8 flow cytometry antibodies, and while we did see a decrease in IRF8 expression with CRISPR deletion, we found that the antibodies tested were not very sensitive. Therefore, we did not include this data in the manuscript. Please see **Reviewer Figure 7** below.

Reviewer Figure 7. Intra-nuclear flow cytometry staining of IRF8. Anti-LeY CAR T cells were cocultured with OVCAR tumor cells for 24 hours. NR4A2/hA₁R or NR4A2/hA₁R/IRF8KO CAR T cells were stained for intra-nuclear expression of IRF8.

9. The IRF8-knockout data are not convincing. While they did not affect cytokine secretion by CAR-T cells in Supp. Figure 8C, it was significantly suppressed in Figure 5I. The authors should provide a more convincing explanation for these different results. Also, CD62L⁺ CD69⁻ cells cannot be defined as memory T cells.

Response: We apologise for the misunderstanding. The original **Figure 5I (new Figure 5L)** refers to A₁R expressing cells and **Supp Figure 8C (New Supp Figure 9C)** refers to control cells. We demonstrate that IRF8 KO reverses the A₁R phenotype (**Figure 5L**), but has no significant effect on cytokine production in control CAR T cells (**New Supp Fig 9C**). This is consistent with the fact that IRF8 was significantly upregulated in A₁R CAR T cells and identified as a key transcription factor for the A₁R phenotype (**Fig5 I-K**), whilst it does not appear to have a role in control CAR T cells. We have further highlighted this point in our revised version of the paper, see lines 631-634.

“CRISPR-mediated deletion of IRF8 (Supp Fig 9B) in control human anti-Lewis Y CAR T cells did not significantly modulate cytokine production or expression of memory and exhaustion markers TCF-1, TIM-3 or PD-1 in unstimulated or tumor stimulated CAR T cells (Supp Fig 9C-E).”

10. Most critically, what is the overall benefit of the strategy of expressing A1R under the NR4A2 promoter compared to the simple knockout of the A2A receptor presented in the previous paper? Since CRISPR/Cas9-mediated gene knock-in is a more complicated procedure, it will not be practically useful unless it has some functional advantage over the knockout of A2A receptor. It will be necessary to directly compare the *in vivo* efficacy, persistence and representative phenotypes of CAR-T cells between control, A1 receptor overexpression, NR4A2-A1R expression, and A2A receptor KO CAR-T cells.

Response: Please see responses to question 3 (Reviewer 2) above. To compare A₁R knock in and A_{2A}R knock-out strategies, we undertook new functional experiments and bulk RNAseq analysis of A₁R expressing vs A_{2A}RKO CAR T cells. Our new data indicate that A₁R has unique effects on CAR T cells independent of A_{2A}R and the two engineering strategies infer distinct transcriptional programs. Interestingly our preliminary analyses revealed synergy in combination A_{2A}RKO and inducible A₁R CAR T cells. We believe that an *in vivo* assessment of this combination approach is beyond the scope of the current paper but we agree with the reviewer that this is an interesting avenue for further exploration.

Reviewer #3 (Remarks to the Author): with expertise in CAR-T cells

I really liked the premise for your paper based on your earlier work. And a really cool advance to limit A1R expression to T cells traversing the TME.

In this article, Sek and coworkers engineer CAR T cells to express A1R, a receptor that opposes the immune-suppressive effects of A2R by dampening levels of the second messenger cAMP. In a comprehensive and extensive analyses (across murine and T cells), constitutive A1R expression drives maximal effector function, albeit at the expense of persistence. This tradeoff can be overcome by limiting A1R expression until CARTs traverse the TME. This innovative advance in site-directed expression led to enhanced anti-tumor function. Compelling mechanistic insights from this study reveal that IRF8 is necessary to drive the enhanced anti-tumor benefit from A1R. This manuscript is well written, articulate, and highly insightful for leveraging T cell differentiation, effector function and persistence following adoptive transfer. It will also appeal to the immunometabolism community at large, as it provides an important tool to skew differentiation/keep it at bay to retain memory cell precursors.

Addressing the following comments will make it a stronger manuscript:

Response: We would like to thank the reviewer for their favourable review of our article.

Major points

1) What is endogenous A₁R expression in naïve, Tcm, and effector T cells? Given its role in effector function, it would be interesting to see if it is induced in effector differentiated cells.

Response: A₁R is not reported to be expressed endogenously in T cells (PMID 11992407) and we also did not observe A₁R mRNA expression in qRT-PCR or RNA-Seq analysis of control CAR T cells and so we were unable to compare expression in different T cell subsets.. We have included a statement to this effect in the revised version of the paper. See lines 666-667.

“Whilst A₁R is not expressed on lymphocytes physiologically, previous studies have suggested a role for A₃R in promoting T cell effector functionality.”

2) How do intracellular cAMP levels change during T cell differentiation (naïve, Tcm, Tem)?

Response: cAMP signalling is a function of extracellular metabolites (predominantly PGE₂ and adenosine) rather than at a T cell differentiation stage. A related relevant question is the expression of the key receptors e.g. A_{2A}R that drive cAMP signaling. Expression of A_{2A}R is increased on activated T cells e.g. T_H1 (PMID 10051547) and elevated on all subsets within the tumor microenvironment (Todd *et al.* Nature Communications 2023). We have referred to this study and other prior literature in the revised paper. See lines 69-71.

How does the anti-tumor function of A₁R-expressing CAR T cells compare to CAR T cells that have A_{2R} deleted (in the same experiment)? It would be good to compare.

Response: Please refer to our responses to questions 3 and 10 (Reviewer 2) above.

How does IRF8 expression change during your chronic stimulation assay? Does its expression coincide with TNFalpha abundance which declines dramatically after the third successive stimulation.

Response: The reviewer has raised an interesting question. We observed that IRF8 transcriptionally increases in the first stimulation and slowly decreases over time as highlighted in **Figure 5K**. Its pattern of expression does coincide with TNF but it is difficult to ascribe a causal relationship.

Additionally, how do intracellular cAMP levels change as T cells undergoing serial stimulation? Is there a dramatic increase after stimulation #3?

Response: Performing cAMP assays on stimulated T cells is technically challenging as the assays needs to be optimised for each cell type used. Please refer to the optimisation data presented in response to reviewer 2 point 1. We therefore do not propose to test this experimentally given that we do not believe that modification of cAMP signaling is a major mechanism for the A₁R mediated effect.

Does IRF8 alter metabolic fitness or is it just linked to differentiation state exclusively? Is it established if adenosine is high in the E0771-Her2 tumor model (Fig 2A)?

Response: While this is a very interesting question, we have not interrogated the role of IRF8 in this context because metabolic pathways were not observed to be significantly modulated by A₁R expression. For example the glycolysis pathway was most significantly associated with the Turquoise gene module (**Figure 5E**), which was not significantly different between control and A₁R expressing CAR T cells (**Supplementary Figure 9A**). However we have added lines 731-732 to the discussion, highlighting a potential role for IRF factors as metabolic stress sensors.

“Additionally, IRF factors have been implicated in metabolic processes as stress sensors, like other factors such as Foxo1, which could be manipulated to further improve CAR-T cell persistence”

Regarding E0771-Her2, it is technically challenging to assess adenosine levels *ex vivo* due to its short half life and we do not have the capacity to do this. Therefore, whilst we cannot answer this question directly, our previous studies have indicated that T cells infiltrating E0771 tumors express high levels of the A_{2A}R and that CAR T cells lacking the A_{2A}R exhibit enhanced anti-tumor activity against E0771-Her2 tumors *in vivo* (Giuffrida, Sek *et al.* 2021, Beavis *et al.* 2017).

Minor Points

It is a pity that transient inhibition of A₁R during expansion (prior to adoptive transfer), didn't enhance anti-tumor function and T cell persistence. How does this reconcile with the corresponding benefit from the induction of A₁R within the tumor?

Response: This is likely explained by the transient nature of A₁R inhibition provided by a pharmacological agent, which is lost upon adoptive transfer. However, NR4A2 regulation means that A₁R is not induced until T cell antigen stimulation by the tumor, therefore overcoming this limitation. We have included a statement in the revised paper to address this point. See lines 738-740 and lines 745-749.

“To overcome this, we utilized a novel CRISPR/Cas9 knock-in strategy to drive A₁R expression specifically at the tumor site, leading to enhanced therapeutic efficacy and comparable engraftment, persistence, and expansion.”

“This highlights the importance of preserving memory but enhancing effector differentiation, and is to our knowledge, the first demonstration of an approach to engineer CAR T cell differentiation states specifically at the tumor site. Importantly this approach can be applied to any receptor or transcription factor that promotes short-term CAR T cell functionality at the expense of long-term persistence.”

Reviewer #4 (Remarks to the Author): with expertise in cancer immunology, adenosine signalling

In this extensive and impressively illustrated study, the authors report that engineering of CAR T cells with A₁R enhances their efficacy against solid tumors *in vitro* and *in vivo*. This is a highly relevant study that could impact therapy of solid tumors with CAR T cells. The latter has not been as satisfactory as expected: only a small percentage of patients treated with CAR T cells achieve CR. This lack of efficacy is attributable to various barriers, but the authors focus on ADO-mediated immune suppression as the major check point for tumor-directed CAR T cells.

Here, they describe “arming” of CAR T cells against immunosuppressive TME by overexpression of A1R, which attenuates suppressive effects of A2AR, a major ADOR in T cells, by inhibition of adenylate cyclase and cAMP accumulation. The authors show that overexpression of A1R and not of A3R suppressed cAMP production and enhanced IFN γ , TNF and granzyme activities in both murine and human CAR T cells. However, constitutive A1R expression resulted in a loss of CAR T cell persistence and reduction in the T stem cell memory (TSCM) subset. So, the authors opted for transient A1R expression by restricting A1R signaling to the tumor site, where ADO is produced. Restriction of A1R to the tumor site was accomplished by using a novel CRISPR knock-in approach under control of the NR4A2 promoter. This CAR T cell modification drove their differentiation and promoted anti-tumor functionality without having any impact on T cell persistence. Thus, the ability to transiently overexpress A1R at the tumor site led to improved therapeutic efficacy by activation of T cells within the tumor but not the spleen. Overexpression of A1R restricted ADO produced by the tumor and reduced ADO mediated suppression of CAR T cells. While the experimental design is rational and experiments were rigorously controlled, there are some concerns. These largely consists of a potential transfer of this methodology to the clinic.

Response: We would like to thank the reviewer for their positive review of our article.

First, knock-in of A1R gene, while feasible, might alter the signaling balance of ADORs in T cells as well as non-immune cells in an unexpected way. Are there any off target effects that the authors observed?

Response: It is not entirely clear what is meant by off target effects in this context. We have further included safety data in the form of mouse weights in **Supp Figure 7B** for a second xenograft mouse tumor and CAR-T model, which infers that there is no off-target toxicity of our approach. With regards to potential off target effects e.g. to non immune cells as a result of our CRISPR engineering strategy, it is important to highlight that the *ex vivo* engineering approach used in our study ensures that edits are specifically made to the CAR T cell population. One concern with regards to CRISPR editing is the potential for off target deletions. To generate an sgRNA with low potential for off target edits we used the COSMID web based tool (PMID 25462530), which indicated no potential off target editing sites for this sgRNA. We agree that for clinical application of this approach further vigorous testing of this sgRNA would be required such as whole genome sequencing as used for sgRNA that have been used to generate products for clinical evaluation. However, this is beyond the scope and resources of this proof of concept preclinical study and even if off target edits were associated with this particularly sgRNA, another could be designed to achieve the same biological effect.

Another concern transient effects of relates to A₁R site-specific overexpression; how will timing of adoptive transfers be orchestrated to achieve optimal therapeutic effects?

Response: In this system CAR T cells upregulate A₁R when they encounter tumor antigen and so the timing of adoptive transfer is inconsequential to this mechanism. However the reviewer eludes to an important question with regards to the optimal magnitude and duration of A₁R expression. This could be optimised through the use of alternative promoters with higher/ lower transgene expression or different kinetics of expression. These investigations are of interest but beyond the scope of the current work.

Also, the authors are overly optimistic about therapeutic efficacy of this therapy targeting but a single suppressive pathway. Although the ADO may be the major checkpoint in some HNCs, it is not the only one. Numerous other immunosuppressive pathways exist, and may differ in functional intensity among patients. Is blockade of one pathway be sufficient to save CAR T cells from tumor-induced suppression??.

Response: We agree with the reviewer that enabling effective treatment of CAR T cells with solid tumors is a significant challenge but this approach demonstrates that it is possible to tune CAR T cell effector differentiation to occur at the “right” time. Importantly, A₁R promotes effector function independently of adenosine mediated suppression, suggesting this approach could be applicable even in situations where adenosine mediated suppression is not prevalent. We agree with the reviewer that there may be other synergistic combinations with A₁R inducible expression that could further enhance CAR-T cell therapy. We are currently exploring other combinations in future studies, including further synergistic targeting of the ADO pathway by A_{2A}R deletion, combination with immune checkpoint inhibitors, and other approaches to promote CAR-T cell memory, persistence and engraftment. To address this point we have included further discussion of this point on line 739-749.

“Several studies have demonstrated the feasibility of favorably controlling CAR T cell differentiation through controlled programming to promote anti-tumor function, most of which have focused on promoting stemness features through inosine [87] or cytokine preconditioning [59, 92, 93] or overexpression of memory associated transcription factors such as FOXO1 [91, 94]. Others have targeted factors that drive or prevent exhaustion such as c-Jun [70], c-Myb [95] or BATF [58]. This highlights the importance of preserving memory but enhancing effector differentiation, and is to our knowledge, the first demonstration of an approach to engineer CAR T cell differentiation states specifically at the tumor site. Importantly this approach can be applied to any receptor or transcription factor that promotes short-term CAR T cell functionality at the expense of long-term persistence.”

The manuscript is clearly written, and the data presentation is excellent.

Response: We thank the reviewer for their positive feedback.

In this article, Sek and coworkers engineer CAR T cells to express A₁R, a receptor that opposes the immune-suppressive effects of A₂R by dampening levels of the second messenger cAMP. In a comprehensive and extensive analyses (across murine and T cells), constitutive A₁R expression drives maximal effector function, albeit at the expense of persistence. This tradeoff can be overcome by limiting A₁R expression until CARTs traverse the TME. This innovative advance in site-directed expression led to enhanced anti-tumor function. Compelling mechanistic insights from this study reveal that IRF8 is necessary to drive the enhanced anti-tumor benefit from A₁R. This manuscript is well written, articulate, and highly insightful for leveraging T cell differentiation, effector function and persistence following adoptive transfer. It will also appeal to the immunometabolism community at large, as it provides an important tool to skew differentiation/keep it at bay to retain memory cell precursors.

Addressing the following comments will make it a stronger manuscript:

Major points

What is endogenous A₁R expression in naïve, Tcm, and effector T cells? Given its role in effector function, it would be interesting to see if it is induced in effector differentiated cells.

How do intracellular cAMP levels change during T cell differentiation (naïve, Tcm, Tem)?

How does the anti-tumor function of A₁R-expressing CAR T cells compare to CAR T cells that have A₂R deleted (in the same experiment)? It would be good to compare.

How does IRF8 expression change during your chronic stimulation assay? Does its expression coincide with TNF α abundance which declines dramatically after the third successive stimulation.

Additionally, how do intracellular cAMP levels change as T cells undergo serial stimulation? Is there a dramatic increase after stimulation #3?

Does IRF8 alter metabolic fitness or is it just linked to differentiation state exclusively?

Is it established if adenosine is high in the E0771-Her2 tumor model (Fig 2A)?

Minor Points

It is a pity that transient inhibition of A₁R during expansion (prior to adoptive transfer), didn't enhance anti-tumor function and T cell persistence. How does this reconcile with the corresponding benefit from the induction of A₁R within the tumor?